# SF-Mamba: Rethinking State Space Model for Vision

## Abstract

The realm of Mamba for vision has been advanced in recent years to strike for the alternatives of Vision Transformers (ViTs) that suffer from the quadratic complexity. While the recurrent scanning mechanism of Mamba offers computational efficiency, it inherently limits non-causal interactions between image patches. Prior works have attempted to address this limitation through various multi-scan strategies; however, these approaches suffer from inefficiencies due to suboptimal scan designs and frequent data rearrangement. Moreover, Mamba exhibits relatively slow computational speed under short token lengths, commonly used in visual tasks. In pursuit of a truly efficient vision encoder, we rethink the scan operation for vision and the computational efficiency of Mamba. To this end, we propose SF-Mamba, a novel visual Mamba with two key proposals: auxiliary patch swapping for encoding bidirectional information flow under an unidirectional scan and batch folding with periodic state reset for advanced GPU parallelism. Extensive experiments on image classification, object detection, and instance and semantic segmentation consistently demonstrate that our proposed SF-Mamba significantly outperforms state-of-the-art baselines while improving throughput across different model sizes. We will release the source code after publication.

## 1 Introduction

The field of deep learning for vision has undergone several architectural shifts, driving progress across a wide range of applications including classification, segmentation, and detection (Deng et al., 2009; Krizhevsky et al., 2012; He et al., 2016; Simonyan & Zisserman, 2014; Ravi et al., 2025; Siméoni et al., 2025; Bolya et al., 2025). More recently, Vision Transformers (ViTs) (Dosovitskiy et al., 2020) have emerged as the dominant paradigm and offer strong flexibility and generalizability compared to convolution-based models by tokenizing images into patches and applying the self-attention mechanism (Vaswani et al., 2017). ViT-based vision models have been widely used for multi-modal learning tasks (Khan et al., 2022; Elharrouss et al., 2025); however, one of the main limitations of ViTs is the quadratic complexity for computing attention in terms of sequence length, which hinders the scalability to high-resolution inputs and large datasets with limited computational resources.

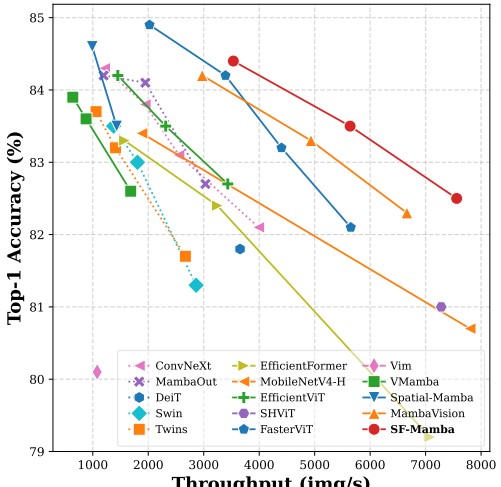

Figure 1: Top-1 accuracy and throughput on ImageNet-1K classification. SF-Mamba offers superior accuracy–throughput trade-offs compared to state-of-the-art architectures.

Mamba (Gu & Dao, 2023) introduces a selective state-space model (SSM), which enables data-dependent flexible token scanning in a left-to-right order, thereby achieving powerful but efficient processing with linear-time complexity. Building on its success, Mamba has been extended to the vision domain, achieving higher accuracy while being efficient in terms of memory cost, FLOPs,

and the number of parameters (Zhu et al., 2024; Liu et al., 2024; Pei et al., 2025). In addition, recent studies suggest that visual Mamba has good transfer learning capabilities comparable to or even surpassing those of ViTs (Yoshimura et al., 2025; Galim et al., 2025), and it has the potential to replace the ViT-based foundation model ecosystem. However, many visual Mamba models suffer from slow processing speeds, especially on low-resolution images, which makes them not truly efficient. One reason is that Mamba adopts a recurrent left-to-right scanning mechanism, which prevents earlier patches from accessing information in future patches. As a result, many visual Mamba methods employ a multi-directional scan strategy, where the input sequence is rearranged and processed from multiple directions (e.g., from top-left to bottom-right, bottom-right to top-left). This allows models to compensate for the inability of standard Mamba to reference future patches and yields strong performance on vision tasks. However, such a rearrangement incurs substantial overhead during both training and inference. In fact, MambaVision (Hatamizadeh & Kautz, 2025) achieves a fast inference by avoiding costly multi-scan and instead rely on attention layers appended after the unidirectional scan. These attention layers enable information to flow backward, allowing earlier tokens to indirectly benefit from later tokens while retaining the efficiency of unidirectional Mamba. Yet, relying solely on attention for backward information flow poses a limitation: backward context can only be injected in deeper layers, leaving shallower layers deprived of future information and potentially restricting the expressiveness of the representation. Another reason why visual Mamba is slow lies in Mamba itself. As reported in the Mamba paper, unless the token length exceeds around 1000–2000, it is slower than Attention. Unless the task involves high-resolution images, the token length of the vision patches typically remains below 1000.

In this paper, we rethink visual Mamba from two perspectives in pursuit of a truly efficient image encoder. The first is the data flow. Instead of using the slower multi-directional scan, we adopt a unidirectional scan. However, this approach lacks future-to-past information flow, which is crucial to generate high-quality features. To address this, we propose an auxiliary patch swapping that enables future-to-past information flow within a unidirectional scan (Sec. 3.2). It introduces two additional tokens mixing the corresponding directional flow, which does not require significant burden compared with existing multi-scan approaches. The second perspective addresses the inefficiency of Mamba when processing short sequences. We attribute this limitation to suboptimal GPU parallelization, and to mitigate it, we introduce batch folding with periodic state reset (Sec. 3.3). This method reshapes batched inputs to maximize GPU thread utilization while preserving independence across sequences, thereby enhancing parallel efficiency. To this end, we propose **SF-Mamba**, which is equipped with the two key innovations of swapping and folding. Extensive experiments on image classification (Fig. 1), object detection, and semantic/instance segmentation demonstrate that SF-Mamba consistently outperforms state-of-the-art baselines while achieving faster inference, paving a new path toward efficient and effective Mamba architectures for vision.

In summary, our main contributions of SF-Mamba are three-fold:

- **Efficient uni-scan for non-causal ordering.** We propose a lightweight mechanism, auxiliary patch swapping, that introduces two learnable auxiliary tokens and a parameter-free swap operation, enabling bidirectional information flow across layers with negligible overhead compared to existing multi-scan approaches.
- **Efficient GPU parallelism for vision tasks.** To address inefficiency in low-resolution vision tasks, we design a batch folding strategy that merges the batch and sequence dimensions, maximizing GPU utilization while preserving the independence of hidden states across sequences. This method can speed up any Mamba-based method especially with short sequence processing.
- **Empirical validation across various tasks.** Experiments on classification, detection, and segmentation show that SF-Mamba outperforms state-of-the-art CNN-, Transformer-, hybrid CNN-Transformer-, and Mamba-based baselines.

## 2 RELATED WORK

**CNNs and Vision Transformers.** Convolutional Neural Networks (CNNs) first led to breakthroughs in large-scale image classification (Deng et al., 2009; Krizhevsky et al., 2012), with deeper networks such as VGG (Simonyan & Zisserman, 2014) and ResNet (He et al., 2016) extending success to segmentation (Long et al., 2015) and detection (Ren et al., 2015). Vision Transformers (ViTs) (Dosovitskiy et al., 2020), inspired by self-attention (Vaswani et al., 2017), have since be-

come the dominant paradigm by effectively modeling long-range dependencies. Follow-up works such as DeiT (d'Ascoli et al., 2021), and Swin Transformer (Liu et al., 2021) improved efficiency and scalability. Models trained on large-scale data are used as a foundation for a variety of tasks (Oquab et al., 2024; Siméoni et al., 2025; Tschannen et al., 2025).

**Visual State Space Models.** To address the quadratic cost of attention, state-space models (SSMs) have emerged as efficient alternatives. Mamba (Gu & Dao, 2023) introduced selective state spaces, enabling linear-time complexity with strong long-range modeling. Inspired by this success, many visual Mamba variants (Zhu et al., 2024; Liu et al., 2024) extended SSMs to visual data.

**Hybrid Architectures in Vision.** Beyond single-paradigm designs, recent studies have demonstrated that hybrid architecture can lead to more efficient encoding. The CNN-Transformer hybrids (Hatamizadeh et al., 2024; Li et al., 2022; Zheng, 2025) leveraged the local feature extraction and inductive biases of CNNs alongside the global context modeling of Transformers. More recently, Mamba–Transformer hybrids (Hatamizadeh & Kautz, 2025) have emerged, combining Mamba's computational efficiency with Transformers' receptive field. The hybrid architecture achieves superior efficiency–performance trade-offs and establishes state-of-the-art vision backbones.

**Causality Constraint of Visual SSMs.** From another perspective, visual SSMs face an inherent challenge: the *causality constraint*, which is also observed in vision-language models (Wang et al., 2025c). Since state-space models process inputs sequentially, each hidden state only depends on the past, preventing access to the global spatial context. Many visual Mamba methods address causality constraints via multi-directional scans. Some approaches like Vim (Zhu et al., 2024) and Mamba-R (Wang et al., 2025a) adopt bi-directional scans, while recent models are based on cross-scan (Liu et al., 2024), which performs bi-directional scan along both horizontal and vertical axes, totaling four directions to better capture image structure. Variants such as GroupMamba (Shaker et al., 2025), MSVMamba (Shi et al., 2024), EfficientVMamba (Shaker et al., 2025), and DefMamba (Liu et al., 2025) enhance cross-scan through zigzag patterns, multi-resolution scan, atrous sampling, or deformable directions. Despite being parameter-efficient, multi-directional scans are slow. Cross-scan based methods are particularly suffer from slow speed due to increased FLOPs from four parallel scans and costly data rearrangement between 2D formats (for 2D convolution) and 1D formats (for scanning). Rearranging tokens for four directions adds further overhead, especially in vertical scans, which involve scattered memory access. While bi-directional scan avoids 2D/1D format switching, it still requires rearranging data for the backward scan and maintaining two parallel paths.

A recent study, Adventurer (Wang et al., 2025b), tackles the causality constraint of Mamba2 (Dao & Gu, 2024) using series bi-directional scans, which alternate scan directions between layers. It also inserts a globally averaged token in every layer to facilitate limited context exchange of series bi-directional scan. While this mechanism only requires single scan in each block, it requires explicit flipping operations with $O(n)$ permutation cost and an additional averaging cost, resulting in reduced throughput.

Several methods are starting to achieve high accuracy with unidirectional scan. Spatial-Mamba (Xiao et al., 2025) uses 2D atrous convolution with a wide receptive field to access future patches, although the 2D/1D format switching degrades the speed. MambaVision (Hatamizadeh & Kautz, 2025) incorporates Attention in later layers to capture global context. However, relying solely on attention for future-to-past information flow might not be optimal.

Furthermore, although previous methods adopt Mamba due to its parameter efficiency and superior accuracy, it remains slower than Attention for token lengths below 1000 to 2000 (Gu & Dao, 2023). These limitations motivate the development of a truly efficient visual Mamba.

## 3 METHOD

### 3.1 PRELIMINARIES

**Mamba State Space Model.** Mamba (Gu & Dao, 2023) is a selective state space model (SSM) that processes a sequence $X = (x_1, \ldots, x_T)$ by recurrently updating a hidden state $h_t$:

$$h_t = A_t h_{t-1} + B_t x_t, \qquad y_t = C_t h_t, \tag{1}$$

where $h_t$ is the hidden state, $y_t$ the output, and $A_t$, $B_t$, $C_t$ are input-dependent matrices.

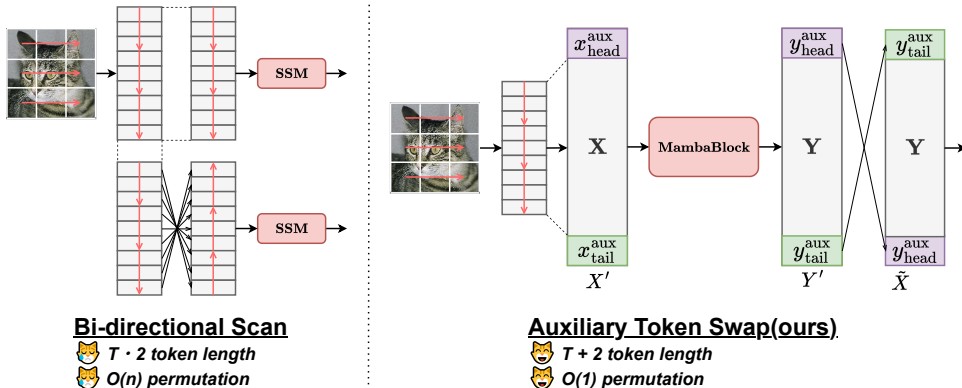

Figure 2: **Future-to-Past Information Routing via Auxiliary Token Swapping.** The left figure illustrates why the commonly used multi-directional scan in visual Mamba fails to achieve high speed, while the right figure presents our proposed solution. We prepend/append learnable auxiliary tokens to the patch sequence $x_{\text{head}}^{\text{aux}}$ and $x_{\text{tail}}^{\text{aux}}$. Within each MambaVision block, the causal selective scan aggregates sequence-wide context into the tail token $y_{\text{tail}}^{\text{aux}}$. A lightweight, parameter-free Swap operation then moves this global summary to the sequence head, yielding $\tilde{X}$ for the next layer such that all patch states are conditioned on global context. It incurs negligible computational overhead while enabling effective global-context propagation across layers.

In the vision setting, an image is divided into $T$ patches, each embedded as $x_t \in \mathbb{R}^D$, forming a sequence $X = (x_1, \ldots, x_T)$. A batch of such sequences is denoted as $X_{\text{in}} \in \mathbb{R}^{B \times T \times D}$, where $B$ is the batch size, $T$ the number of patches (sequence length), and $D$ the embedding dimension. In this case, Mamba can be viewed as a mapping $f_\theta : \mathbb{R}^{T \times D} \to \mathbb{R}^{T \times D}$ that applies the recurrence in eq. (1) to patch sequences.

Using a *parallel scan* algorithm, this recursive operation is efficiently computed. Specifically, Mamba uses a warp scan function (NVIDIA, 2025a) implemented in the CUDA backend, which enables high-speed parallel scan by allowing multiple threads to share data through the fast SRAM memory of the GPU. Since this warp scan function operates in groups of 32 threads, each sequence must be processed using at least 32 threads.

**Mamba-Transformer Hybrid Architecture.** We employ a Mamba-Transformer hybrid architecture, because previous studies indicate that the hybrid architecture achieves promising efficiency. In other words, we employ MambaVision (Hatamizadeh & Kautz, 2025) architecture as a macro level. It uses a four-stage hierarchical design. The first two stages are CNN-based and serve as a kind of deep patch embedding. The latter two stages consist of several Mamba blocks followed by several Attention Blocks. MambaVision accelerates processing by adopting a simple unidirectional scan. However, due to the *causality constraint*, it cannot reference future patches from past ones, so future-to-past information flow relies on subsequent Attention blocks. Detailed structure and formulation of the MambaVision-based blocks are provided in Appendix C.1.

## 3.2 RETHINKING VISUAL SSM FROM DATA FLOW PERSPECTIVE

**Future-to-Past Information Routing via Auxiliary Token Swapping.** Since image patches do not exhibit a strict causal ordering, restricting Mamba blocks to a unidirectional scan can be limiting: tokens in earlier regions (e.g., the top-left) cannot directly access information from later regions (e.g., the bottom-right), which hinders representation learning. While multi-scan approaches such as bidirectional or cross-scan alleviate this issue, they require repeated reordering of the data, which introduces substantial computational overhead and complicates implementation. Hence, we propose a future-to-past information flow with minimal additional cost by introducing two *auxiliary tokens*.

At the first Mamba block in each stage, the two auxiliary tokens, $x_{\text{head}}^{\text{aux},1}$ and $x_{\text{tail}}^{\text{aux},1}$, are initialized as data-dependent values (i.e., $x_{\text{head}}^{\text{aux},1} = x_{\text{tail}}^{\text{aux},1} = avg(X)$, where $avg(\ )$ averages the sequential

$$\mathbf{B} \times \mathbf{D} \times \mathbf{T} \qquad \mathbf{B_1} \times \mathbf{D} \times (\mathbf{B_2} \cdot \mathbf{T})$$

**Periodic State Reset Trick**

1: **for** $t = 1$ **to** $B_2 \cdot T$ **do**
2:     **if** $t \bmod T = 0$ **then**
3:         $A_t \leftarrow 0$
4:     $h_t \leftarrow A_t h_{t-1} + B_t x_t$
5:     $y_t \leftarrow C_t h_t$

Figure 3: **Batch folding with periodic state reset.** (**Left**) An input tensor of shape $[B, D, T]$ is reshaped into $[B_1, D, (B_2 \cdot T)]$, concatenating $B_2$ short sequences into a longer one. This reshaping mixes hidden states across batches. (**Right**) To avoid information leakage, we reset the recurrence every $T$ steps. Since $h_t \leftarrow A_t h_{t-1} + B_t x_t$, setting $A_t = 0$ at boundaries is equivalent to re-initializing the hidden state. In contrast, $B_t$ (input projection) and $C_t$ (output projection) operate locally and therefore remain unchanged.

dimension). The tokens are then concatenated at both ends of the input $X$ for the first Mamba block:

$$X' = (x_{\text{head}}^{\text{aux},1}, \ x_1, \ldots, x_T, \ x_{\text{tail}}^{\text{aux},1}). \tag{2}$$

After processed by the $i$-th Mamba block, we swap the two tokens for the input of the next Mamba block (see Fig. 2):

$$x_{\text{head}}^{\text{aux},i+1} = y_{\text{tail}}^{\text{aux},i}, \quad x_{\text{tail}}^{\text{aux},i+1} = y_{\text{head}}^{\text{aux},i}, \tag{3}$$

where $y_{\text{head}}^{\text{aux},i}$ and $y_{\text{tail}}^{\text{aux},i}$ are the output tokens with respect to $x_{\text{head}}^{\text{aux},i}$ and $x_{\text{tail}}^{\text{aux},i}$. By training this architecture, we expect that $y_{\text{tail}}^{\text{aux},i}$ extracts the necessary information from all tokens in the $i$-th layer, and $y_{\text{head}}^{\text{aux},i}$ serves as a feature that determines how $y_{\text{tail}}^{\text{aux},i+1}$ should be extracted in the next layer. Then, by swapping as shown in Eq. 3, the patch tokens of the next layer ($x_1, x_2, ..., X_T$) can refer to $x_{\text{head}}^{\text{aux},i+1}$, which contains features from all positions, allowing future-to-past information routing. This intended operation is natural for the selective scan SSM and does not disrupt the original mechanism, which selectively extracts the necessary information as $y_t$ from hidden states that span from $t = 0$ to $t = t$. Similarly, we expect it selectively extracts the necessary information as $y_{\text{tail}}^{\text{aux},i}$ from hidden states that span from $t = 0$ to $t = T$.

In contrast to multi-scan strategies, our approach does not rely on multiple parallel paths or global token rearrangements. Instead, it swaps only two tokens within the sequence, introducing negligible computational overhead.

## 3.3 RETHINKING VISUAL SSM FROM COMPUTATIONAL PERSPECTIVE

**Batch Folding with Periodic State Reset.** We identify that Mamba's inefficiency in low-resolution vision tasks arises from the warp-scan implementation, which achieves high throughput by utilizing 32 GPU threads per sequence (Sec. 3.1). In vision models, however, the number of patches (i.e., sequence length) is relatively small (e.g., 196 and 49 for MambaVision Stage 3 and 4), making the allocation of 32 threads per sequence highly underutilized and inefficient. To address this, we propose a batch folding strategy that reshapes the input by merging the batch dimension into the sequence dimension (Fig. 3, left). This improves parallel efficiency in scenarios with many short sequences while preserving the correctness of the computation. Let $Z \in \mathbb{R}^{B \times D \times T}$ denote the batched tokens before entering the SSM. We reshape $Z$ into

$$Z' \in \mathbb{R}^{B_1 \times D \times (B_2 \cdot T)}, \quad B = B_1 \cdot B_2, \tag{4}$$

which concatenates $B_2$ short sequences into one longer sequence. This operation is a bijective permutation of indices, so the original tensor can be exactly recovered. Intuitively, this extends the effective sequence length in a pseudo manner, allowing the parallel scan to operate more efficiently by reducing kernel launch overhead and reducing inefficient use of memory bandwidth.

However, this reshaping mixes hidden states across different sequences. To preserve independence, we effectively use and improve a computational trick implemented in vLLM software (Kwon et al., 2023), which was originally devised for Mamba inference in LLMs to handle multiple sequences

of varying lengths without padding. Our trick for preserving the dependence of the folded data named *periodic state reset trick* is as follows (Fig. 3, right). In every $T$ step, we set $A_t = 0$, which removes dependence on $h_{t-1}$ and resets the hidden state. Then, all the hidden states become identical to those without batch folding. By unfolding the output, the output becomes equivalent to that obtained without applying batch folding. Note that $B_t$ and $C_t$ act only on the current input and hidden state, respectively, and thus do not require resetting. Since it only resets A, there is only a minimal increase in processing time.

**Adaptive $B_1$.** In batch folding, it is not optimal to increase the virtual sequence size indefinitely. The ideal ratio between $B_1$ and $B_2$ is determined in a complex manner based on factors such as batch size $B$, number of input tokens $T$, model dimension $D$, state dimension $S$, and the number of threads used when invoking CUDA. Therefore, we precompute and store combinations of (B, D, L, S), along with the optimal $B_1/B$ ratio, in a coarse-grained 4-dimensional lookup table ($LUT$). At runtime, we retrieve the optimal $B_1$ value from this $LUT$ as follows:

$$B_1 = f(B, B \cdot LUT(B, D, S, L)), \tag{5}$$

where $f(a, b)$ is a function that returns a divisor of $a$, which is closest to $b$.

**1-D Depthwise Convolution for Batch Folded Data.** Although batch folding improves the speed of the SSM component, the reshaping operation in Eq. 4 introduces a slowdown. To mitigate this, we apply the transformation in Eq. 4 only at the initial Mamba block of each stage, and then continue computation using the batch-folded tensor shape. Since the Linear and LayerNorm layers operate per token, they do not pose any issues. However, the 1D depthwise convolution in the Mamba block presents a challenge. To address this, we implement a convolution that supports batch-folded data, ensuring that no convolution occurs across the boundaries between T sequences. In other words, our convolution CUDA kernel performs implicit padding at the boundary of each T sequence.

## 4 EXPERIMENTS

We conduct comprehensive experiments to evaluate SF-Mamba across three fundamental computer vision tasks: image classification, semantic segmentation, object detection with instance segmentation (Appendix D.3). Our experimental setup follows the protocols established by previous works (Liu et al., 2024; Xiao et al., 2025; Hatamizadeh & Kautz, 2025) to ensure fair comparisons. We evaluate three model variants (T/S/B) with different scales to analyze the accuracy-throughput trade-offs. For all downstream tasks, we use models pre-trained on ImageNet-1K as backbones. Detailed training configurations and hyperparameters are provided in Appendix B.

### 4.1 IMAGE CLASSIFICATION

**Experimental Setup.** We first evaluate our models on image classification task using ImageNet-1K (Deng et al., 2009), which contains 1.28M training images and 50K validation images across 1,000 categories. Models are trained from scratch for 300 epochs following prior works (Hatamizadeh & Kautz, 2025; Liu et al., 2024). Throughput is measured on a single NVIDIA A100 GPU with a batch size of 128 (see Appendix B.4 for details).

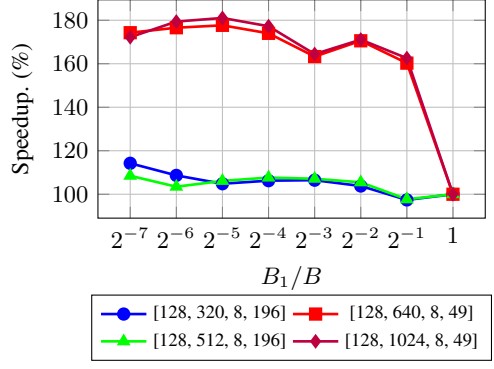

Figure 4: How much we can speedup the SSM calculation by changing $B_1$. The four configurations of [batch size, dimension, state dimension, sequence length] are exact settings for ours-T stage 3, ours-T stage 4, ours-B stage 3, ours-B stage 4.

**Results.** As shown in Fig. 1, SF-Mamba achieves superior efficiency-accuracy trade-offs with consistent improvements across all model scales (T/S/B variants) compared to existing architectures including CNN-based models (ConvNeXt (Liu et al., 2022), MambaOut (Yu & Wang, 2025)), Transformer-based models (DeiT (Touvron et al., 2021), Swin (Liu et al., 2021), Twins (Chu et al., 2021)), hybrid CNN-Transformer ar-

chitectures (EfficientFormer (Li et al., 2022), Ef-
ficientVit (Cai et al., 2023), MobileNetV4-H-M (Qin et al., 2024), SHViT (Yun & Ro, 2024), Faster-
ViT (Hatamizadeh et al., 2024)), and recent Mamba-based models (Vim (Zhu et al., 2024), VMamba
(Liu et al., 2024), Spatial-Mamba (Xiao et al., 2025), MambaVision (Hatamizadeh & Kautz, 2025)).
Note that some models in Fig. 1 have non-224×224 input resolution (see Appendix B. Tab. 4
provides a detailed comparison of models evaluated at the standard 224×224 resolution.

**Analysis.** To analyze the effect of *batch folding with periodic state reset*, the speed of the SSM
kernel part is measured, as shown in Fig. 4. A clear speedup of 110% to 180% is observed when the
batch dimension is virtually shifted into the sequential dimension. This improvement is especially
significant when the sequential length is short. The reason lies in the CUDA parallel scan algorithm,
which plays a crucial role in Mamba's speedup. This algorithm requires at least 32 threads per
sequence, and when the sequence is short, the overhead of allocating 32 threads becomes substantial.
By virtually extending the sequence length, we can utilize the allocated threads more efficiently,
leading to a significant performance boost.

Table 1: The computational speedup achieved by our method

| impl. opt. | BFold | $B_1$ | conv | img/s |
|:---:|:---:|:---:|:---:|:---:|
| | | | | 6662 |
| ✓ | | | | 6989 |
| ✓ | ✓ | 1 | ✓ | 7601 |
| ✓ | ✓ | 4 | ✓ | 7641 |
| ✓ | ✓ | adaptive | | 7279 |
| ✓ | ✓ | adaptive | ✓ | **7685** |

Next, we evaluate how much our proposed method can improve the overall model speed, as shown
in Tab. 1. Our baseline is MambaVision-T, and we measure the degree of speed improvement from
it. First, we improve the implementation and observe a speedup, which we denote as *impl. opt*. This
improvement primarily stems from the SSM CUDA kernel, which we rewrote based on the Mamba
SSM CUDA kernel to suit our method. Building upon this, our *batch folding with periodic state
reset (BFold)* technique achieves a significant speedup. Furthermore, our *adaptive $B_1$* approach,
which adaptively adjusts $B_1$ according to input and weight sizes, enables further improvements in
inference speed. Since MambaVision is a hybrid model that combines Attention and Mamba, it goes
without saying that it does not achieve the same level of speed improvement as described in Fig. 4.
However, it still delivers significant performance gains compared to the baseline.

Table 2: The effectiveness of auxiliary token swapping and its ablation results.

| swap | auxiliary token | discard timing | IN1K acc. | ADE20K IoU | img/s |
|:---:|:---:|:---:|:---:|:---:|:---:|
| | | | 82.2 | 46.0 | 7645 |
| | learnable | before attn | 82.1 | 46.2 | 7613 |
| ✓ | learnable | before attn | 82.3 | 46.5 | 7585 |
| ✓ | data-dependent | before attn | 82.4 | 46.8 | 7602 |
| ✓ | data-dependent | after 1st attn | **82.5** | **47.2** | 7600 |
| ✓ | data-dependent | after attn | 82.4 | 46.6 | 7597 |

Tab. 2 presents an ablation study on *auxiliary token swapping*. The swapping improves performance
with only a minimal impact on inference speed. Simply adding learnable tokens without performing
swapping degrades performance, indicating that the improvement does not come from the increased
flexibility provided by the additional tokens, but rather from the bidirectional information flow en-
abled by swapping. Looking at Fig. 7 in the Appendix, we can clearly see that it indeed achieves
substantial bi-directional information propagation. Initializing auxiliary tokens as globally averaged
data-dependent values proves more effective than employing a learnable token, which is commonly
used as a class (Dosovitskiy et al., 2020; Zhu et al., 2024). In addition, Initializing this token with
global features may allow subsequent layers to effectively acquire the global information needed for
the next layer. As for where to discard this token, the most efficient approach is to remove it after
the first attention layer.

Table 3: Comparison of effective scan designs. The "series bi-scan + gap token" follows Adventurer (Wang et al., 2025b), where a global token obtained via global average pooling is used. In parallel bi-scan, "cat" splits the input along channels, applies bi-scan, and concatenates the results, while "add" duplicates the input, applies bi-scan, and sums the outputs.

| macro-arch. | MambaVision-T | | | | MambaVision-T w/o Attention | | | |
| scan | Params | MACs | img/s | acc. | Params | MACs | img/s | acc. |
|---|---|---|---|---|---|---|---|---|
| uni-scan | 31.8M | 4.4G | 6979 | 82.2 | 29.4M | 4.2G | 6238 | 80.2 |
| series bi-scan | 31.8M | 4.4G | 6911 | 82.3 | 29.4M | 4.2G | 6113 | 80.4 |
| series bi-scan+gap token (Adventurer) | 31.8M | 4.5G | 6856 | 82.3 | 29.4M | 4.3G | 6027 | 80.7 |
| parallel bi-scan (cat) | 31.8M | 4.4G | 6834 | 82.2 | 29.4M | 4.2G | 5987 | 80.8 |
| parallel bi-scan (add) | 31.9M | 4.5G | 6235 | 82.3 | 29.7M | 4.3G | 5138 | 81.1 |
| parallel bi-scan (add) (Vim block) | 33.5M | 4.6G | 4612 | 82.4 | 32.8M | 4.6G | 3256 | **81.7** |
| uni-scan + swap (ours) | 31.8M | 4.5G | **6926** | **82.5** | 29.4M | 4.3G | **6171** | 81.0 |
| uni-scan + swap (ours) + Bfold | 31.8M | 4.5G | 7600 | 82.5 | 29.4M | 4.3G | 7306 | 81.0 |

Table 4: Detailed comparison of image classification performance on ImageNet-1K. All models in this table are evaluated with $224 \times 224$ input resolution.

| Model | Token Mixer | Params | MACs | Throughput (img/s) | Top-1 Acc (%) |
|---|---|---|---|---|---|
| ConvNeXt-T | Conv | 29M | 4.5G | 3990 | 82.1 |
| MambaOut-T | Conv | 27M | 4.5G | 3031 | 82.7 |
| Swin-T | Attn | 29M | 4.5G | 2863 | 81.3 |
| Twins-S | Attn | 24M | 2.9G | 2669 | 81.7 |
| EfficientFormer-L3 | Pool + Attn | 31M | 3.9G | 3246 | 82.4 |
| FasterViT-0 | Conv + Attn | 31M | 3.3G | 5651 | 82.1 |
| Vim-S | Conv + SSM | 26M | 5.3G | 1079 | 80.1 |
| VMamba-T | Conv + SSM | 30M | 4.9G | 1684 | 82.6 |
| Spatial-Mamba-T | Conv + SSM | 27M | 4.5G | 1430 | 83.5 |
| MambaVision-T | Conv + SSM + Attn | 32M | 4.4G | 6662 | 82.3 |
| **SF-Mamba-T** | Conv + SSM + Attn | 32M | 4.5G | **7600** | 82.5 |
| ConvNeXt-S | Conv | 50M | 8.7G | 2552 | 83.1 |
| MambaOut-S | Conv | 49M | 9.0G | 1948 | 84.1 |
| Swin-S | Attn | 50M | 8.7G | 1805 | 83.0 |
| Twins-B | Attn | 56M | 8.6G | 1409 | 83.2 |
| FasterViT-1 | Conv + Attn | 53M | 5.3G | 4402 | 83.2 |
| EfficientViT-B3 | Conv + Attn | 49M | 4.0G | 2315 | 83.5 |
| VMamba-S | Conv + SSM | 50M | 8.7G | 879 | 83.6 |
| Spatial-Mamba-S | Conv + SSM | 43M | 7.1G | 990 | 84.6 |
| MambaVision-S | Conv + SSM + Attn | 50M | 7.5G | 4933 | 83.3 |
| **SF-Mamba-S** | Conv + SSM + Attn | 50M | 7.6G | **5639** | 83.5 |
| ConvNeXt-B | Conv | 89M | 15.4G | 1943 | 83.8 |
| MambaOut-B | Conv | 85M | 15.9G | 1195 | 84.2 |
| Swin-B | Attn | 88M | 15.4G | 1377 | 83.5 |
| Twins-L | Attn | 99M | 15.1G | 1059 | 83.7 |
| EfficientFormer-L7 | Pool + Attn | 82M | 10.2G | 1573 | 83.3 |
| FasterViT-2 | Conv + Attn | 76M | 8.7G | 3392 | 84.2 |
| VMamba-B | Conv + SSM | 89M | 15.4G | 640 | 83.9 |
| Spatial-Mamba-B | Conv + SSM | 96M | 15.8G | 670 | 85.3 |
| MambaVision-B | Conv + SSM + Attn | 98M | 15.0G | 2974 | 84.2 |
| **SF-Mamba-B** | Conv + SSM + Attn | 98M | 15.1G | **3534** | 84.4 |

Tab. 3 demonstrates which scan method is most efficient. Here, we evaluate with two macro-architectures. One is MambaVision-T, and the other is an architecture in which all attention blocks in MambaVision-T are replaced into Mamba blocks. The parallel bidirectional scan (bi-scan) requires twice the SSM computation cost due to its parallel nature, resulting in inefficient computation. Even in parallel bi-scan (cat), which halves the channel dimension to align MACs, the speed is slow due to the tensor rearrangement cost.

In contrast, the series bidirectional scan flips the token sequence at each layer, with odd-numbered blocks scanning in the forward direction and even-numbered blocks scanning in the reverse direction. This design allows for the creation of global features without increasing the number of FLOPs. However, the accuracy improvement is not as significant as expected. We hypothesize that DropOut (Huang et al., 2016), which is effective in preventing overfitting and gradient vanishing, may not be compatible with the series bidirectional scan architecture, which has an asymmetric structure across layers. To this end, Adventurer (Wang et al., 2025b) style model does not achieve high accuracy, although the introduced global averaged token actually improves from the normal series bi-scan setting. Also, the flipping operation needed for the bi-scan the block incurs the speed with an $O(n)$ permutation cost.

On the other hand, our *auxiliary token swapping* only swaps two tokens, minimizing the slowdown while achieving comparable or superior accuracy. The fact that the swapping improves a lot from unidirectional scan with the Mamba only architecture indicates that it allows future-to-past token information flow with the swapping, thereby facilitating the creation of better features (See Fig. 7).

## 4.2 SEMANTIC SEGMENTATION

Table 5: Semantic segmentation performance on ADE20K dataset using UperNet. We compare with Swin Transformer (Liu et al., 2021), Focal Transformer (Yang et al., 2021), and MambaVision (Hatamizadeh & Kautz, 2025). All models are trained at a resolution of 512×512 while FLOPs are calculated with an input size of 2048×512. Frames per second (FPS) are measured with a batch size of 1. SF-Mamba♣ uses a windowed attention to save computational cost.

| Backbone | Tiny-size | | | | Small-size | | | | Base-size | | | |
| --- | --- | --- | --- | --- | --- | --- | --- | --- | --- | --- | --- | --- |
| | Para | FLOPs | mIoU | fps | Para | FLOPs | mIoU | fps | Para | FLOPs | mIoU | fps |
| Swin | 60M | 945G | 44.5 | 40.0 | 81M | 1038G | 47.6 | 25.7 | 121M | 1188G | 48.1 | 25.4 |
| Focal | 62M | 998G | 45.8 | 38.9 | 85M | 1130G | 48.0 | 24.0 | 126M | 1354G | 49.0 | 23.4 |
| MambaVision | 62M | 1085G | 46.0 | 45.0 | 81M | 1166G | 48.2 | 40.9 | 130M | 1520G | 49.1 | 37.3 |
| **SF-Mamba** | 62M | 1085G | **47.2** | 47.9 | 81M | 1166G | **48.5** | 45.4 | 130M | 1520G | **50.1** | 42.6 |
| **SF-Mamba♣** | 62M | 950G | 46.5 | **48.7** | 81M | 1014G | 48.1 | **47.3** | 130M | 1180G | 49.1 | **42.7** |

**Experimental Setup.** We evaluate on ADE20K (Zhou et al., 2017) using UperNet (Xiao et al., 2018) as the segmentation framework. This task requires assigning a semantic class label to each pixel in the image across 150 categories. Models are trained with 512×512 crop resolution following standard protocols (Liu et al., 2024; Xiao et al., 2025). Performance is measured by mean Intersection over Union (mIoU) (Csurka et al., 2013).

**Results.** Tab. 5 summarizes the semantic segmentation performance on ADE20K. Segmentation, unlike image classification, requires both fine-grained pixel-level boundary detection and global structural understanding to accurately identify object classes. Therefore, enabling the Mamba block to incorporate future patch information through state swapping significantly improves the accuracy compared to the baseline MambaVision. During inference, the model processes images at a resolution of 512×2048, which differs from the training resolution. To accommodate this discrepancy, both

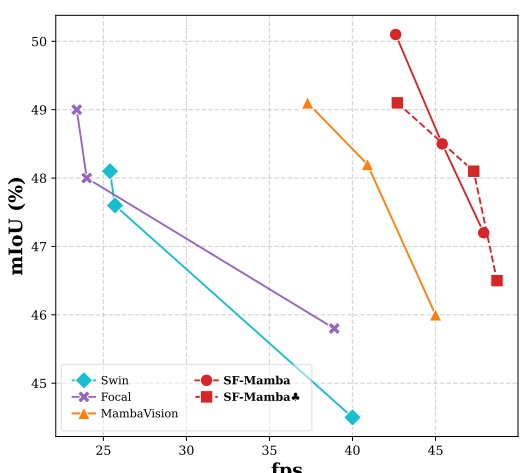

Figure 5: **Throughput–accuracy trade-off on ADE20K.** The x-axis denotes frames per second with batch size 1 setting (higher is better), and the y-axis shows mIoU (higher is better). SF-Mamba variants lie on the Pareto front.

Mamba and Attention are implemented to process per windowed region, where the window size matches the training image dimensions (see Appendix C.3 for details). Therefore, although the inference speed is measured with a batch size of 1, the model still benefits from batch folding based

on the number of windows, resulting in an improved speed. Our base size model is faster compared to the Tiny versions of Swin and Focal Transformer, while achieving over 4 points higher mIoU. This fact can be clearly understood if we visualize the accuracy-speed trade-off as shown in Fig. 5. The best trade-off among recent visual backbones indicates that the proposed auxiliary patch swapping improves both efficiency and generalization capability, offering superior accuracy per unit of computational cost.

The SF-Mamba♣ configuration in Table 5 adopts more granular window attention to reduce the quadratic computational cost of Attention by dividing the image into smaller windows. Meanwhile, Mamba is not affected by long tokens. So, we do not use a window size smaller than the training image size for the Mamba blocks to capture global context, resulting in better efficiency in terms of FLOPs. More details can be found in Appendix D.4. Moreover, Table 12 in Appendix indicates that much larger input images benefit a large speed gain in SF-Mamba♣ configuration.

## 5 CONCLUSION

In this paper, we rethink the recently effective visual Mamba approach from two perspectives. The first is an efficient scanning method for vision tasks. Previous studies have addressed the causality constraint of SSM by introducing multiple scan directions, but this comes with a significant drop in inference speed. To overcome this, we propose auxiliary token swapping, which enables future-to-past information flow without sacrificing inference speed, thereby achieving efficient scanning. The second perspective investigates why Mamba tends to be slow in image processing. We identified the bottleneck and proposed batch folding, a method that virtually extends the sequence length while keeping the identical SSM output, resulting in faster processing without accuracy drop. SF-Mamba, a novel Mamba-based framework with these proposals, achieves a superior accuracy-speed trade-off compared to existing methods. Although the latter technique may not provide benefits during inference with batch size = 1, training typically uses batch size > 1, so the speed-up advantage is expected in most training scenarios. Moreover, even with a batch size of 1, Mamba-based approaches–such as those employing local windows as in our segmentation experiments or multi-directional scan–result in an effective batch size larger than 1 for the SSM, thereby allowing for performance acceleration. We believe that this work will advance the development of efficient and effective image recognition models.

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

## A  BROADER IMPACTS

Our work aims to improve the computational efficiency of state-space models for vision tasks, which has potential benefits for both large-scale and resource-constrained deployment scenarios. The proposed swapping and batch-folding mechanisms offer improved throughput at low resolution and ultra-high resolution (See Table 12). This may reduce training and inference costs in high-resolution applications such as medical imaging, aerial monitoring, and robotics.

Although we were unable to include evaluations on edge GPUs or mobile hardware in this submission, prior studies (Wei, 2025; NVIDIA, 2025b) have shown that Mamba kernels can be deployed on devices such as NVIDIA Jetson and iOS through optimized runtimes (e.g., TensorRT, mobile accelerators). Our method should be adaptable to these platforms since the same selective scan is used in our core algorithm. Enabling efficient state-space model inference on edge devices may broaden access to low-power real-time vision systems, but also calls for careful consideration of responsible deployment in safety-critical or privacy-sensitive contexts.

## B  EXPERIMENTAL SETUP DETAILS

### B.1  IMAGE CLASSIFICATION

We train our SF-Mamba variants (Tiny/Small/Base) on the ImageNet-1K dataset (Deng et al., 2009), which contains 1.28M training images and 50K validation images across 1,000 categories. Following the protocol of MambaVision (Hatamizadeh & Kautz, 2025), we adopt standard data augmentation (RandAugment, Mixup, CutMix) and regularization (Label Smoothing, Stochastic Depth). The detailed hyperparameter settings are summarized in Table 6.

Table 6: Training configurations for SF-Mamba variants on ImageNet-1K. All models are trained for 300 epochs following the MambaVision configuration (Hatamizadeh & Kautz, 2025).

| Configuration | SF-Mamba-T | SF-Mamba-S | SF-Mamba-B |
|---|---|---|---|
| Optimizer | LAMB (You et al., 2019) | LAMB | LAMB |
| Base learning rate | 5e-3 | 5e-3 | 5e-3 |
| Learning rate schedule | Cosine | Cosine | Cosine |
| Warmup epochs | 20 | 20 | 35 |
| Warmup learning rate | 1e-6 | 1e-6 | 1e-6 |
| Minimum learning rate | 5e-6 | 5e-6 | 5e-6 |
| Weight decay | 0.05 | 0.05 | 0.075 |
| Optimizer momentum ($\beta_1$) | 0.9 | 0.9 | 0.9 |
| Optimizer momentum ($\beta_2$) | 0.999 | 0.999 | 0.999 |
| Optimizer epsilon | 1e-8 | 1e-8 | 1e-8 |
| Gradient clipping (norm) | 5.0 | 5.0 | 5.0 |
| Total epochs | 300 | 300 | 300 |
| Batch size (total) | 4,096 | 4,096 | 4,096 |
| Input resolution | 224×224 | 224×224 | 224×224 |
| Mixup alpha | 0.8 | 0.8 | 0.8 |
| CutMix alpha | 1.0 | 1.0 | 1.0 |
| RandAug (Cubuk et al., 2020) | rand-m9-mstd0.5 | rand-m9-mstd0.5 | rand-m9-mstd0.5 |
| Label smoothing | 0.1 | 0.1 | 0.1 |
| Random erasing prob. | 0.25 | 0.25 | 0.25 |
| Model EMA | ✓ | ✓ | ✓ |
| EMA decay | 0.9998 | 0.9998 | 0.9998 |
| Mixed precision (AMP) | ✓ | ✓ | ✓ |

### B.2  OBJECT DETECTION AND INSTANCE SEGMENTATION

For MS COCO (Lin et al., 2014), we use Cascade Mask R-CNN (Cai & Vasconcelos, 2019b) implemented in MMDetection (Chen et al., 2019). All backbones are initialized from ImageNet-1K

pre-training. We use AdamW as the optimizer and adopt the commonly used $3\times$ training schedule. The batch size is 16. Further details follow MambaVision (Hatamizadeh & Kautz, 2025).

### B.3 SEMANTIC SEGMENTATION

For ADE20K (Zhou et al., 2017), we use UperNet (Xiao et al., 2018) implemented in MMSeg-mentation (Contributors, 2020). Backbones are initialized with ImageNet-1K pre-training. We use AdamW as the optimizer with batch size 16. A polynomial learning rate decay schedule is applied, consistent with MambaVision (Hatamizadeh & Kautz, 2025).

### B.4 THROUGHPUT MEASUREMENT

We measure throughput on an NVIDIA A100 40GB GPU with a batch size of 128 and input images of size $224\times224$ using automatic mixed precision, following established protocols (Hatamizadeh et al., 2024; Hatamizadeh & Kautz, 2025). Note that some models in Fig. 1 have non-$224\times224$ input resolution following the original setup ( e.g. MobileNetV4-H-M (Qin et al., 2024): $256\times256$, MobileNetV4-H-L: $384\times384$, EfficientViT-B2(r256) (Cai et al., 2023): $256\times256$, EfficientViT-B3(r288): $288\times288$, SHViT (Yun & Ro, 2024): $384\times384$). Our software environment consists of CUDA 12.4, cuDNN 9, and PyTorch 2.6.0. To ensure a fair comparison, we measure the through-put of all previous methods under the same experimental settings. We report the speed of the faster memory format between channel last and channel first. The reported throughput values are the medians over 500 inference runs, and for ours-T, the variation across 10 trials was $7600 \pm 11$.

## C IMPLEMENTATION DETAILS

### C.1 MACRO-ARCHITECTURE

MambaVision (Hatamizadeh & Kautz, 2025) combining Attention and Mamba, is a state-of-the-art model as a macro-level structure for vision tasks, excelling in speed, performance, and scalabil-ity. Therefore, our macro-architecture follows MambaVision with a four-stage hierarchical design. Given an input image $I \in \mathbb{R}^{H \times W \times 3}$, the stem and successive stages transform the resolution and channel dimension as follows:

$$
\begin{aligned}
&\text{Stage 1:} \quad I \xrightarrow{\text{Stem + ConvBlock} \times N_1} \tfrac{H}{4} \times \tfrac{W}{4} \times D, \\
&\text{Stage 2:} \quad \tfrac{H}{4} \times \tfrac{W}{4} \times D \xrightarrow{\text{Downsample + ConvBlock} \times N_2} \tfrac{H}{8} \times \tfrac{W}{8} \times 2D, \\
&\text{Stage 3:} \quad \tfrac{H}{8} \times \tfrac{W}{8} \times 2D \xrightarrow{\text{Downsample + } \textit{MambaBlock} \times N_3/2 + \textit{AttenBlock} \times N_3/2} \tfrac{H}{16} \times \tfrac{W}{16} \times 4D, \quad (6) \\
&\text{Stage 4:} \quad \tfrac{H}{16} \times \tfrac{W}{16} \times 4D \xrightarrow{\text{Downsample + } \textit{MambaBlock} \times N_4/2 + \textit{AttenBlock} \times N_4/2} \tfrac{H}{32} \times \tfrac{W}{32} \times 8D, \\
&\text{Classifier:} \quad \tfrac{H}{32} \times \tfrac{W}{32} \times 8D \xrightarrow{\text{Global AvgPool + Linear}} \mathbb{R}^{\#\text{classes}}.
\end{aligned}
$$

where $N_i$ denotes the number of blocks to apply sequentially. In Stage 3 and 4, $N_i$ Mamba Blocks are applied followed by $N_i$ Attention Blocks. The Mamba Block consists of a MambaVision Mixer and an MLP. The MambaVision Mixer takes input as a patch sequence $X_{\text{in}} \in \mathbb{R}^{B \times T \times D}$ and pro-cesses it through two parallel branches: a selective SSM and a local convolutional path. Formally,

$$
\begin{aligned}
X_1 &= \text{SSM}\Big(\sigma(\text{Conv}(\text{Linear}_{D \to D}(X_{\text{in}})))\Big), \quad X_2 = \sigma(\text{Conv}(\text{Linear}_{D \to D}(X_{\text{in}}))), \\
Y &= \text{Linear}_{2D \to D}(\text{Concat}(X_1, X_2)).
\end{aligned} \quad (7)
$$

Here, $\sigma$ is a SiLU activation (Elfwing et al., 2018), $\text{Conv}$ is a 1-D depthwise convolution, and $\text{SSM}(\cdot)$ denotes the SSM with selective scan $h_t = A_t h_{t-1} + B_t x_t$, $y_t = C_t h_t$. The two paths are fused and projected back to dimension $D$, yielding the output $Y$. Unlike many visual Mamba methods (Liu et al., 2024; Wang et al., 2025a), the MambaVision Mixer accelerates processing by adopting a simple unidirectional scan. However, due to the *causality constraint*, it cannot reference future patches from past ones, so future-to-past information flow relies on subsequent Attention blocks.

## C.2 Implementation Optimization for Faster Inference

As indicated by "impl. opt" in Tab. **??**(a), we apply several implementation level optimizations not written in the method section to accelerate inference. The details of these implementation optimizations are listed below:

- **Removal of unused row-dimension chunking in the Mamba SSM kernel**: As with VMamba (Liu et al., 2024), we remove the unused row (channel) dimension chunking feature from the Mamba SSM kernel. This allows more intermediate variables to be handled as float values rather than float arrays, resulting in improved speed.

- **Suppressing hidden state output during inference**: The Mamba SSM kernel is modified so that it does not output hidden states except during training. Since hidden states are only needed for backpropagation, avoiding their output during inference reduces unnecessary memory write time.

- **Replacing linear layers with pointwise 1D Convolution**: As with VMamba, we replace the linear layer that output the $\Delta t$ tensor with pointwise 1D convolutions. This reduces unnecessary tensor rearrangement.

- **Auxiliary token swapping with a Triton CUDA kernel**: Although the computational cost of the swapping is not significant, swapping data at non-contiguous positions is needed, especially with the batch folded data. Converting this process to a Triton CUDA kernel improves throughput slightly by about 40 img/s for ours-T and about 10 imag/s for ours-B, although we can use none-Triton swapping for simplicity.

## C.3 Segmentation and Object Detection

In image classification, we followed the MambaVision meta-architecture. However, for object detection and instance segmentation on the COCO dataset (Lin et al., 2014), and semantic segmentation on ADE20K (Zhou et al., 2017), we make some modifications. The reason is that processing high-resolution images with Attention incurs significant computational cost. Based on our analysis, it appears that MambaVision mistakenly omits the computational cost of Attention in terms of FLOPs for the COCO and ADE20K tasks. Therefore, the FLOPs values for MambaVision in our table differ from those reported in the original paper.

To address this, we made two improvements to create a more lightweight model architecture. The first is to remove excessive padding regions. In MambaVision, large padding areas are added in both the Mamba Block and the Attention Block to serve as additional computation regions, thereby improving accuracy. Although it leads to a degradation in accuracy, we reduce computational cost by removing these extra padding regions and lowering the resolution in Stage 3 and Stage 4 (e.g. Stage 3: 112×112 to 84×84, Stage 4: 56×56 to 42×42 for COCO).

The second improvement is the use of windowed Attention. Since Attention has a quadratic cost with respect to token length, we reduce computational cost by applying local windowed Attention to Stage 3, which has a long sequence length. This also results in a slight drop in performance.

After applying these changes, our model architecture is as follows: The stem layer, Stage 1, and Stage 2 are convolution-based and process the input image directly. Stage 3 processes features padded to a resolution of 84×84 for COCO and 64×64 for ADE20K. Stage 4 processes images at 42×42 for COCO and 32×32 for ADE20K. Padding is necessary because these tasks require handling images with various aspect ratios and resolutions. For task-specific decoders–Cascade Mask RCNN (Cai & Vasconcelos, 2019a) (for COCO) and UperNet (Xiao et al., 2018) (for ADE20K)– the padding regions are removed before input. When using windowed Attention, the window size in Stage 3 is set to 42×42 for COCO and 32×32 for ADE20K. During training on ADE20K, the model is trained with an input resolution of 512×512, whereas during evaluation it needs to process resolutions up to 2048×512. Therefore, in Stage 3 and Stage 4 during the evaluation, both the Mamba and Attention blocks handle feature maps larger than 64×64 or 32×32 by dividing them into windowed patches for processing.

# D  ADDITIONAL EXPERIMENTS

## D.1  PRELIMINARY EVALUATION ON MULTI-DIRECTIONAL SCAN COST

We measure how much existing multi-directional scan methods affect throughput. As representative examples of multi-directional scan, we experiment with bi-directional scan (Zhu et al., 2024) and cross-scan (Liu et al., 2024), which are commonly used as the basis for many scanning methods Wang et al. (2025a); Shi et al. (2024); Pei et al. (2025). To accurately identify the causes of performance degradation, we conduct the following three simple experiments. The first experiment measures the throughput using the original model structure as proposed, which includes multi-directional scan. The second experiment measures the throughput of a model whose scan directions are replaced with forward-only scans. The last experiment measures the throughput of a model where all non-forward scan directions are removed from the original model. The difference between the first and second experiments reflects the time spent on reordering tokens, which is required by multi-directional scans. The difference between the second and third experiments indicates the time cost of performing scans in parallel.

Fig. 6 shows how much of the total inference time is occupied by these components. Surprisingly, we found that token reordering, which is not reflected in FLOPs, accounts for 5–8% of the total processing time in models using multi-directional scan. Furthermore, performing parallel multi-directional scans consumes an additional 28–42% of processing time, which means that the accuracy gains of multi-directional scan must outweigh this cost. In the case of VMamba, the time spent rearranging the data between 2D and 1D formats is additionally hidden under the "others" category.

Our method is also included in the table as a reference. Direct comparison is difficult since our model uses Attention too and the proportion of Mamba blocks is relatively small. However, auxiliary token swapping in our method results in negligible processing time. As a result, in addition to the effectiveness of batch folding, our model is significantly faster, although all three models have nearly identical FLOPs.

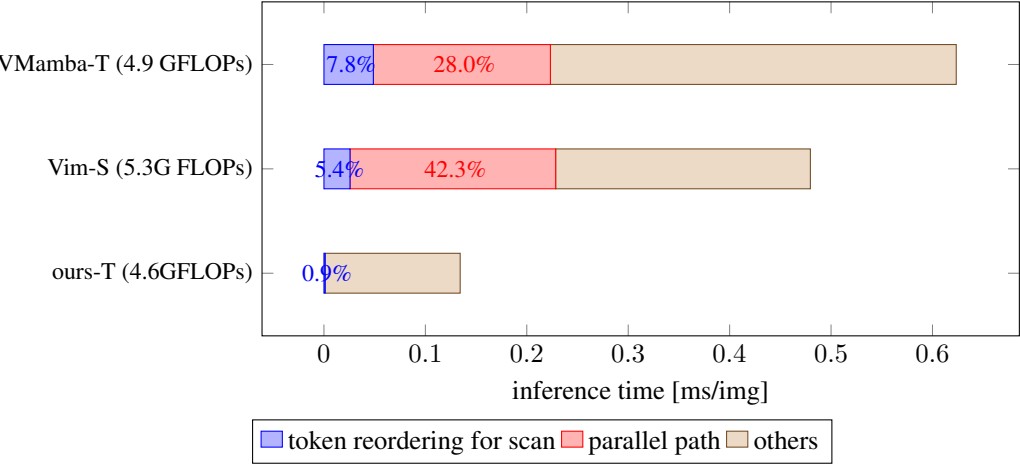

Figure 6: Computational cost of multi-directional scan. This includes the time required to reorder tokens for scanning from multiple directions, and the additional processing time incurred by setting up parallel paths.

## D.2  EFFECTIVE RECEPTIVE FIELD ANALYSIS

To better understand how our model captures spatial dependencies, we conduct an Effective Receptive Field (ERF) analysis (Luo et al., 2016; Ding et al., 2022) following the methodology of Liu et al. (2024). The ERF is computed by measuring the squared gradient of the output features with respect to the center pixel, which highlights the regions most influential for each prediction. Fig. 7 shows the ERF corresponding to the layers up to the Stage 3 Mamba blocks of tiny sized models. So, the ERF for only Convolution and two Mamba blocks is shown. MambaVision uses a simple unidirectional

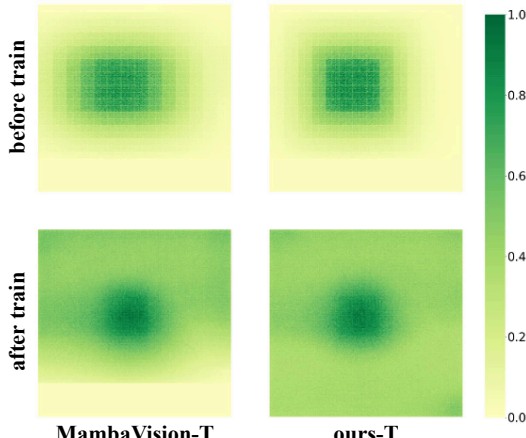

Figure 7: Effective Receptive Field (ERF) comparison. This ERF corresponds to the layers up to the Stage 3 Mamba blocks. MambaVision uses a simple unidirectional scan, which prevents it from accessing future tokens (i.e., the lower part of the image) beyond what can be captured by convolution. In contrast, SF-Mamba leverages auxiliary token swapping, allowing it to account for both past and future tokens with similar strength. Since the auxiliary token swapping information propagation follows the mechanism of SSM, it can be effectively achieved with just two tokens.

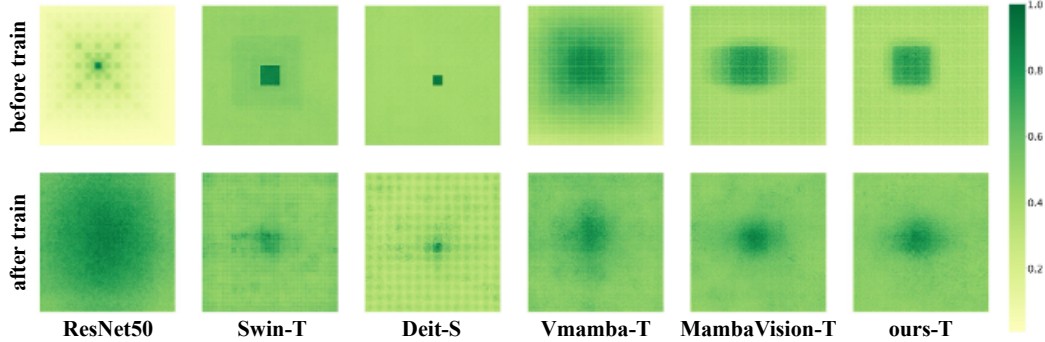

Figure 8: Effective Receptive Field (ERF) comparison of the entire model. SF-Mamba achieves globally distributed ERFs with reduced computational complexity.

scan, which prevents it from accessing future tokens (i.e., the lower part of the image) beyond what can be captured by convolution. In contrast, SF-Mamba leverages auxiliary token swapping, allow- ing it to account for both past and future tokens with similar strength. Since the information propaga- tion of auxiliary token swapping follows the mechanism of SSM, it can be effectively achieved with only two tokens. Fig. 8 compares the ERFs of entire models. SF-Mamba leverages auxiliary patch swapping to facilitate a global receptive field while maintaining high throughput. Unlike attention- based architectures whose cost scales quadratically with the sequence length, SF-Mamba avoids this overhead thanks to its state-space formulation. This demonstrates that SF-Mamba achieves global context modeling with improved computational efficiency.

### D.3 OBJECT DETECTION AND INSTANCE SEGMENTATION

**Experimental Setup.** We evaluate on MS COCO 2017 (Lin et al., 2014) using Cascade Mask R- CNN (Cai & Vasconcelos, 2019a) as the detection framework. The task involves localizing objects with bounding boxes (detection) and predicting pixel-level masks for each instance (segmentation). We follow the standard $3\times$ training schedule. We report both bounding-box average precision (AP) and mask AP metrics following the COCO evaluation protocol (Lin et al., 2014).

Table 7: Object detection and instance segmentation performance on MS COCO dataset using Cascade Mask R-CNN. All models are trained with $3\times$ schedule at $1280\times800$ resolution. We compare with ConvNeXt (Liu et al., 2022), Swin Transformer (Liu et al., 2021), and MambaVision (Hatamizadeh & Kautz, 2025). SF-Mamba♣ uses a windowed Attention (no window for Mamba blocks) to save computational cost.

| Backbone | Params | FLOPs | fps | $AP^b$ | $AP^b_{50}$ | $AP^b_{75}$ | $AP^m$ | $AP^m_{50}$ | $AP^m_{75}$ |
|---|---|---|---|---|---|---|---|---|---|
| Swin-T | 86M | 745G | 26.3 | 50.4 | 69.2 | 54.7 | 43.7 | 66.6 | 47.3 |
| ConvNeXt-T | 86M | 741G | **32.1** | 50.4 | 69.1 | 54.8 | 43.7 | 66.5 | 47.3 |
| MambaVision-T | 89M | 1118G | 19.4 | **51.1** | 70.0 | 55.6 | **44.3** | 67.3 | 47.9 |
| **SF-Mamba-T** | 89M | 741G | 27.8 | 51.0 | 69.9 | 55.3 | 44.2 | 67.1 | 48.0 |
| **SF-Mamba♣-T** | 89M | **659G** | 28.3 | 50.9 | 69.9 | 55.0 | 44.1 | 66.9 | 47.7 |
| Swin-S | 107M | 838G | 18.7 | 51.9 | 70.7 | 56.3 | 45.0 | 68.2 | 48.8 |
| ConvNeXt-S | 108M | 827G | 28.0 | 51.9 | 70.8 | 56.5 | 45.0 | 68.4 | 49.1 |
| MambaVision-S | 107M | 1192G | 20.5 | 52.3 | 71.1 | 56.7 | 45.2 | 68.5 | 48.9 |
| **SF-Mamba-S** | 107M | 817G | 28.7 | **52.4** | 71.1 | 56.7 | **45.4** | 68.5 | 49.1 |
| **SF-Mamba♣-S** | 107M | **731G** | **28.8** | 52.1 | 71.0 | 56.4 | 45.2 | 68.4 | 48.8 |
| Swin-B | 145M | 982G | 18.6 | 51.9 | 70.5 | 56.4 | 45.0 | 68.1 | 48.9 |
| ConvNeXt-B | 146M | **964G** | 26.0 | 52.7 | 71.3 | 57.2 | 45.6 | 68.9 | 49.5 |
| MambaVision-B | 155M | 3000G | 16.4 | **52.8** | 71.3 | 57.2 | 45.7 | 68.7 | 49.4 |
| **SF-Mamba-B** | 155M | 1185G | 26.8 | **52.8** | 71.3 | 57.2 | **45.8** | 68.9 | 49.4 |
| **SF-Mamba♣-B** | 155M | 992G | **27.6** | 52.6 | 71.3 | 57.2 | 45.7 | 69.0 | 49.2 |

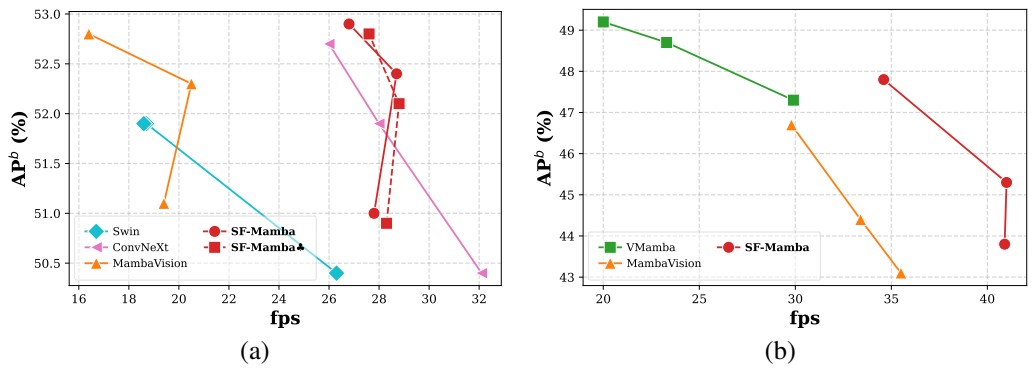

(a)                                                         (b)

Figure 9: Accuracy-speed trade-off on MS COCO using (a) Casecade Mask-RCNN Cai & Vasconcelos (2019b) and (b) Mask R-CNN He et al. (2017). Casecade Mask-RCNN is trained with the 3x schedule while Mask-RCNN is trained with the 1x schedule. In MambaVision and in our model built on its macro-architecture, the tiny and small variants show a reversal in speed. This is because the tiny model has three Attention layers in stage 3 and two in stage 4, whereas the small model has two Attention layers in stage 3 and three in stage 4. When high-resolution images are used as input, Attention computation becomes the bottleneck. Consequently, even though the tiny model has fewer parameters, its throughput becomes lower due to the larger number of Attention layers in stage 3.

**Results.** Tab. 7 presents the object detection and instance segmentation results on MS COCO. Our approach again achieves improvements in both accuracy and efficiency over the baseline and also outperforms the Swin and Focal Transformer, indicating its general applicability. As discussed in Appendix D.4, by removing the extensive padding region used in the baseline and replacing global attention with window-based attention, we achieve substantial efficiency gains at the expense of some accuracy. For example, SF-Mamba♣-S, utilizing window Attention to save computational cost, has smaller FLOPs than the tiny size models while achieving 1.0 and 0.9 points higher $AP^b$ and $AP^m$. Thanks to the introduction of state swapping, we attain performance comparable to or even surpassing the baseline. The clear improvement over the existing methods can be seen in Fig. 9(a).

### D.4 Evaluation of Excessive Padding and Windowed Attention in Segmentation and Detection Tasks

Table 8: The impact of excessive padding regions and the use of windowed Attention in terms of computational cost and accuracy. With the excessive padding setting, Stage 3 uses large padding sizes — 112×112 for COCO and 64×64 for ADE20K. In contrast, the *w/o pad* configuration minimizes padding to match the training image sizes, resulting in 84×84 for COCO and 32×32 for ADE20K. Regarding local windows: A3 refers to the windowed Attention in Stage 3 and M3 refers to the windowed Mamba in Stage 3. The configurations used for our models, SF-Mamba and SF-Mamba♣ , are highlighted. For ADE20K, since large images are processed during testing, we retain the large padding.

| arch. | w/o pad | A3 | A4 | M3 | M4 | ADE20K FLOPs | mIoU | COCO FLOPs | mAP$^b$ | mAP$^m$ |
|---|---|---|---|---|---|---|---|---|---|---|
| w/o swap | | | | | | 1085G | 46.0 | 1118G | 50.9 | 44.1 |
| w/ swap | | | | | | 1085G | 47.2 | 1118G | 51.2 | 44.5 |
| w/ swap | ✓ | | | | | 950G | 45.8 | 741G | 51.0 | 44.2 |
| w/ swap | | 4 | | | | 950G | 46.2 | 741G | 50.9 | 44.0 |
| w/ swap | ✓ | 4 | | | | 942G | 45.6 | 736G | 50.9 | 44.0 |
| w/ swap | ✓ | 4 | 4 | | | 941G | 45.6 | 649G | 50.5 | 44.0 |
| w/ swap | ✓ | 4 | 4 | 4 | 4 | 941G | 45.4 | 649G | 50.3 | 43.9 |

As outlined in Appendix C.3, Tab. 8 presents an ablation study on the impact of excessive padding regions and the use of windowed Attention in terms of computational cost and accuracy. Introducing excessive padding enables the model to utilize additional spatial regions for additional computational area, which leads to a modest accuracy gain. However, due to the quadratic scaling of Attention with respect to token length, this improvement comes at a substantial computational cost.

We further examine the effect of applying Attention or Mamba within local windows. This consistently resulted in accuracy degradation, suggesting that both mechanisms are effective in capturing long-range dependencies. It is an advantage over convolutional approaches. Despite the drop in accuracy, windowed Attention significantly reduces computational overhead. In contrast, Mamba maintains linear complexity with respect to sequence length, which means that windowing does not reduce its computational cost. Based on these findings, our SF-Mamba♣ applies windowing exclusively to Attention, while utilizing Mamba for global modeling. Since windowed Attention restricts complete future-to-past token information flow, our auxiliary token swapping mechanism plays a critical role in enabling bidirectional context propagation. For high-resolution inputs, the benefits of Mamba over Attention become even more pronounced. This indicates that increasing the use of Mamba may further enhance performance in high-resolution segmentation and detection tasks.

### D.5 Object Detection and Instance Segmentation with Other Detection Heads

**Experimental Setup.** To evaluate the generality of our method on downstream tasks and compare it with other existing image encoders, we also perform experiments on the COCO dataset (Lin et al., 2014) using Faster R-CNN (Ren et al., 2015) and Mask R-CNN (He et al., 2017) detection heads. Following prior work (Radosavovic et al., 2020; Liu et al., 2024), we train all models with the 1×

schedule (12 epochs). The baselines which we compare are RegNetX (Radosavovic et al., 2020), VMamba (Liu et al., 2024), and MambaVision (Hatamizadeh & Kautz, 2025). For MambaVision, we disable excessive padding and instead use the same minimal padding strategy as our method to compare with a comparable computational cost.

**Results.** We first present the Faster R-CNN results in Table 9. Under the aligned experimental settings and matched computational cost, our method achieves a substantial performance improvement over MambaVision.

The results with Mask-RCNN is shown in Table 10 and Fig. 9, showing consistent accuracy and speed improvements over our baseline again. Although our method is weaker in accuracy compared to VMamba within the same categories (T, S, or B), when comparing VMamba-**T** and SF-Mamba-**B**, SF-Mamba-B surpasses VMamba-T in both speed and accuracy, clearly demonstrating a superior performance–throughput trade-off. The Fig. 9(b) should be easy to understand the trade-off.

Table 9: Detection on COCO dataset using Faster R-CNN. The models are trained with 1x schedule (12 epoch).

| Backbone | AP box | fps |
|---|---|---|
| RegNetX-3.2GF | 39.9 | 31.6 |
| MambaVision-T | 42.4 | 37.2 |
| SF-Mamba-T | **43.2** | **41.7** |
| MambaVision-S | 43.9 | 37.2 |
| SF-Mamba-S | **44.9** | 40.8 |
| MambaVision-B | 46.2 | 30.0 |
| SF-Mamba-B | **47.6** | **34.8** |

Table 10: Detection and instance segmentation on COCO dataset using Mask R-CNN. The models are trained with 1x schedule (12 epoch).

| Backbone | $AP_{box}$ | $AP_{mask}$ | FPS |
|---|---|---|---|
| VMamba-T | 47.3 | 42.7 | 29.9 |
| MambaVision-T | 43.1 | 40.0 | 35.5 |
| SF-Mamba-T | 43.8 | 40.3 | 40.9 |
| VMamba-S | 48.7 | 43.7 | 23.3 |
| MambaVision-S | 44.4 | 41.0 | 33.4 |
| SF-Mamba-S | 45.3 | 41.5 | 41.0 |
| VMamba-B | 49.2 | 44.1 | 20.0 |
| MambaVision-B | 46.7 | 42.8 | 29.8 |
| SF-Mamba-B | 47.8 | 43.5 | 34.6 |

## D.6 APPLICABILITY TO OTHER VISION MAMBA VARIANTS

Our two core contributions—auxiliary patch swapping and batch folding with periodic reset—can be integrated into other visual Mamba variants. To demonstrate this, we add experiments on Vim (Zhu et al., 2024) architecture as shown in Table 11.

When we remove the modules responsible for the inverse-direction scan in the Vim-S architecture and make the model uni-directional, the accuracy drops, but the speed improves significantly. By introducing the proposed auxiliary token swapping, we can recover most of the lost accuracy while preserving the improved speed. Furthermore, unlike the MambaVision macro-architecture, Vim makes extensive use of Mamba blocks, which allows our batch folding to yield a substantial speed improvement. The resulting model achieves a speed similar to Vim-T but with substantially better accuracy than Vim-T, demonstrating that our method offers a clearly superior accuracy–throughput trade-off.

Our method has proven effective for architectures such as MambaVision and Vim but, there are also limitations on which Visual Mamba models it can be applied to. For example, it is difficult to adapt

our approach to architectures like VMamba (Liu et al., 2024), which incorporate 2D convolutions inside the Mamba module. This is because these models must convert the data back into a 2D format at every layer, but once auxiliary tokens are added, the data no longer conform to the original 2D format. In addition, they cannot process data in the batch-folded representation; instead, they must reconvert the data into the non-batch-folded format at every layer, which degrades the speed-up.

Table 11: Comparison on Vim-S macro-architecture. To match the parameter count, we increase the channel dimension from 384 to 400 in the uni-scan model.

| size | scan | Params | MACs | img/s | acc. |
|---|---|---|---|---|---|
| S | parallel-bi scan (Vim) | 26M | 5.3G | 1079 | **80.3** |
| S | uni-scan | 26M | 4.9G | 1639 | 79.3 |
| S | uni-scan + swap | 26M | 5.0G | 1614 | 80.1 |
| S | uni-scan + swap + Bfold | 26M | 5.0G | **2022** | 80.1 |
| T | parallel-bi scan (Vim) | 7M | 1.5G | 2094 | 76.3 |

### D.7 THROUGHPUT EVALUATION UNDER VARIOUS SCENARIOS

Here, we evaluate throughput across a variety of scenarios.

**Higher Input Resolutions.** A throughput comparison at higher input resolutions is shown in 12. These results show that our proposed improvements preserve their benefits even as resolution scales. The strategy using windowed Attention while using Mamba globally (SF-Mamba) remains particularly robust under extremely large inputs. This may reduce training and inference costs in high-resolution applications such as medical imaging, aerial monitoring, and robotics.

Table 12: Throughput (images/s) for different models and resolutions. OOM denotes out-of-memory with an A100 40GB GPU. We use a windowed Attention with a $32\times32$ size for SF-Mamba♣, the same as the ADE20K setup. The "-" for SF-Mamba♣ means that the feature sizes of both stage 3 and 4 are less than $32\times32$, so the same with SF-Mamba.

| Model (batch size) | 224 | 448 | 896 | 1792 | 3584 |
|---|---|---|---|---|---|
| VMamba-T (bs=32) | 1384 | 402 | 107 | 5 | OOM |
| FasterViT-0 (bs=32) | 1415 | 1400 | 418 | 99 | OOM |
| MambaVision-T (bs=32) | 3770 | 1578 | 324 | 50 | 5 |
| SF-Mamba-T (bs=32) | 3962 | 1777 | 397 | 61 | 6 |
| SF-Mamba♣-T (bs=32) | - | - | 427 | 105 | 27 |
| VMamba-T (bs=1) | 62 | 62 | 62 | 24 | 5 |
| FasterViT-0 (bs=1) | 44 | 43 | 43 | 43 | 12 |
| MambaVision-T (bs=1) | 119 | 121 | 120 | 48 | 5 |
| SF-Mamba-T (bs=1) | 126 | 126 | 125 | 54 | 6 |
| SF-Mamba♣-T (bs=1) | - | - | 120 | 89 | 26 |

**Different Batch Sizes.** The throughput measured under different batch sizes is summarized in Table 13. Although auxiliary token swapping introduces a slight increase in computational cost, the results show that throughput consistently improves thanks to batch folding and other optimizations.

Table 13: Throughput with various batch sizes

| Arch. | 1 | 32 | 128 | 256 | 512 | 1024 |
|---|---|---|---|---|---|---|
| MambaVision-T | 119 | 3770 | 6662 | 7025 | 7134 | 7271 |
| ours-T | 126 | 3962 | 7600 | 7801 | 8009 | 8190 |
| MambaVision-B | 96 | 2798 | 2974 | 3128 | 3176 | 3206 |
| ours-B | 98 | 3168 | 3534 | 3592 | 3641 | 3685 |

## D.8 CONTRIBUTION OF ATTENTION AND MAMBA

Here, We analyze the contribution of Attention and Mamba as shown in Table 14. These results show that while Attention provides beneficial bidirectional information flow, it alone is not sufficient to match the full hybrid model. In contrast, SSM alone lags behind, but incorporating the swapping mechanism yields a clear improvement. The best performance is achieved only when both components–Attention and SSM (with swap)–are present. This supports our claim that token swapping plays a complementary role to Attention rather than replacing it. Thanks to our batch folding with periodic reset and auxiliary-token swapping, we can leverage Mamba to achieve improvements in the accuracy–speed trade-off even for low-resolution inputs.

Table 14: The contribution of Attention and Mamba.

| Arch. | Params | img/s | acc. |
|---|---|---|---|
| Attention only | 34.2M | **7803** | 82.3% |
| SSM only | 29.4M | 6238 | 80.2% |
| SSM only (w/ our Bfold and swap) | 29.4M | 7306 | 81.0% |
| Hybrid | 31.8M | 6979 | 82.2% |
| Hybrid (w/ our Bfold and swap) | 31.8M | 7600 | **82.5%** |

