# OpenReview forum: "SF-Mamba: Rethinking State Space Model for Vision"
_ICLR.cc/2026/Conference — Submitted to ICLR 2026_

### Official Review · Reviewer_ypap · 2025-10-15

**Soundness:** 2
**Presentation:** 2
**Contribution:** 2
**Rating:** 4
**Confidence:** 5

**Summary:**

This paper introduces SF-Mamba, a vision model designed to improve the efficiency and performance of State Space Models (SSMs) for visual tasks. It proposes two main contributions: Auxiliary Patch Swapping to enable bidirectional information flow in a unidirectional scan, and Batch Folding with Periodic State Reset to improve GPU parallelism for short sequences common in vision.

**Strengths:**

The authors correctly identify that visual Mamba models are often slow not just due to scan strategies but because of suboptimal GPU utilization on the short sequences common in vision tasks. The "Batch Folding" technique is a clever, hardware-aware solution that provides significant speedups.

**Weaknesses:**

**Limited Novelty**

Overlap with Adventurer [1]: The core idea of swapping auxiliary tokens to enable bidirectional information flow is conceptually almost identical to the flip operation between consecutive blocks proposed in Adventurer (CVPR 2025). Adventurer also uses this technique to address the causality constraint in unidirectional visual models. The absence of any discussion, citation, or empirical comparison to this highly relevant prior work is a major oversight and significantly weakens the paper's claim to novelty.

Batch Folding as an Engineering Optimization: While the Batch Folding technique is shown to be effective, it functions more as a low-level implementation optimization or an "engineering trick" rather than a novel academic contribution. Its impact is valuable for performance but its conceptual depth is limited for a top-tier conference paper.


**Questionable Motivation**

The paper's motivation rests on the claim that multi-directional scans are inherently slow, but this is not convincingly proven and contradicts previous findings.

Contradiction with VMamba: The authors claim inefficiency in multi-directional scans, but this conclusion is challenged by the original VMamba paper, which demonstrated that a multi-directional scan could achieve comparable throughput to a single-directional one. This discrepancy is not addressed.


Unfair Ablation Study (Table 3): The ablation study in Table 3, which aims to show the slowness of multi-scan methods, is methodologically flawed. The comparisons do not keep the parameters and MACs constant across different scan methods. For example, a "parallel bi-scan" would naturally have a higher computational cost unless channel dimensions are halved to ensure a fair comparison, which does not appear to be the case here. This invalidates the conclusion that the proposed uni-scan with swapping is inherently faster than a properly configured multi-scan architecture.


**Writing and Paper Structure**

An extensive amount of space in the main paper is used to describe the MambaVision macro-architecture. This detailed background could be moved to the appendix. This would free up valuable space to provide a more thorough analysis of the results on downstream tasks like COCO and ADE20K, which currently feel rushed and lack sufficient detail.


**Marginal Performance on Downstream Tasks**

While SF-Mamba excels on ImageNet classification, its advantages diminish significantly on more complex downstream tasks. The performance improvements over the MambaVision baseline are minimal for semantic segmentation on ADE20K.
More critically, on the MS COCO object detection and instance segmentation tasks, some SF-Mamba configurations underperform the MambaVision baseline they are supposed to improve upon.


[1]. Causal Image Modeling for Efficient Visual Understanding. CVPR2025.

**Questions:**

Please check weaknesses.

---

> ### Author Response · Authors · 2025-11-27
> **Reply to ypap (W1)**
>
> We thank the reviewer for the detailed and thoughtful feedback. We appreciate the recognition of our GPU-level optimization insights and the effectiveness of batch folding. Below we address the main concerns regarding novelty, fairness, and motivation.
>
> ##  Relation to Adventurer (W1)
>
> We thank the reviewer for pointing out the similar work, Adventurer [1]. While both methods aim to accelerate inference by reducing scan parallelism, our approach achieves a more efficient pipeline. Adventurer uses a globally average-pooled token to achieve some degree of bidirectional information flow, and compensates for the remaining deficiency by employing a series bidirectional scan. In contrast, we use a uni-directional scan and realize bidirectional information flow through our auxiliary token swapping.
>
> In terms of speed, our method has two advantages. First, using a uni-directional scan eliminates the computation cost associated with flipping tokens. Second, auxiliary token swapping can be computed in parallel with scanning the other tokens, whereas the global average pooling operation in Adventurer must be executed independently of other computations, which slows down the speed.
>
> Regarding accuracy, as discussed in the experimental section, the series bidirectional scan appears to interact poorly with DropPath, leading to lower accuracy compared with our auxiliary token swapping approach. Furthermore, our auxiliary tokens, like regular patch tokens, can selectively extract the necessary information, allowing them to achieve higher performance than a global average token. In fact, we additionally evaluated Adventurer’s approach, but as shown below, both its speed and accuracy were degraded compared to ours. (Note that the original Adventurer paper uses Mamba-2, and we did not have sufficient time to adapt our method to Mamba-2.)
>
> **[Comparison with Adventurer]**
> | Attention in arch. | w/ |  |  | w/o |  |  |
> |--------|--------|------|------|------|------|------|
> | scan| MACs | img/s | acc. | MACs | img/s | acc. |
> | uni-scan | 4.4G | 6979 | 82.2 | 4.2G | 6238 | 80.2 |
> | series bi-scan | 4.4G | 6911 | 82.3 | 4.2G | 6113 | 80.4 |
> | series bi-scan+gap token (Adventurer) | 4.5G | 6856 | 82.3 | 4.3G | 6027 | 80.7 |
> | uni-scan + swap (ours) | 4.5G | 6926 | **82.5** | 4.3G | 6171 | **81.0** |
> | uni-scan + swap (ours) + Bfold (ours) | 4.5G | **7600** | **82.5** | 4.3G | **7306** | **81.0** |
>
>
> ## On the Nature of Batch Folding (W1)
> We acknowledge that Batch Folding is a computational method, but emphasize that it is conceptually motivated by the state-space model’s recurrent dependency. This is not merely a kernel trick, but a structural design that bridges algorithmic causality and GPU-level scheduling, improving both training and inference throughput.
>
> In our view, proposing a DNN architecture and proposing a new computation method differ only in whether the coding effort lies in the upper-layer logic or the lower-layer logic. Both changes the way the input data processed. That difference alone should not determine whether the work is engineering or research. Rather, we believe the difference comes from the intention and contribution: if the work simply improves performance through incremental refinements, it is engineering; if it identifies a concrete limitation of existing methods and introduces a novel solution to that problem, then it should be research. We believe our method is the latter.

---

> > ### Author Response · Authors · 2025-11-27
> > **Reply to ypap (W2, W3, W4)**
> >
> > ## Motivation and Multi-directional Scan Efficiency (W2)
> > We agree that parallel bidirectional scan should be tried with half channel dimension to align MACs. We tried and the result is shown as "parallel bi-scan 1/2" below.
> >
> > **[Comparison with Adventurer]**
> > | Attention in arch. | w/ |  |  | w/o |  |  |
> > |--------|--------|------|------|------|------|------|
> > | scan| MACs | img/s | acc. | MACs | img/s | acc. |
> > | series bi-scan | 4.4G | 6911 | 82.3 | 4.2G | 6113 | 80.4 |
> > | parallel bi-scan 1/2| 4.4G | 6834 | 82.2 | 4.2G | 5987 | 80.8 |
> > | parallel bi-scan | 4.5G | 6235 | 82.3 | 4.3G | 5138 | **81.1** |
> > | uni-scan + swap (ours) | 4.5G | **6926** | **82.5** | 4.3G | **6171** | 81.0 |
> >
> > It shows that even under equal computational cost, our uni-scan with swapping remains a little faster due to reduced reshaping cost. Moreover, if dimension is halved, the accuracy becomes lower than auxiliary token swapping. These results prove the superiority of our methods.  Note that "parallel bi-scan 1/2" is slower than "series bi-scan" because "parallel bi-scan 1/2" has to flip the tokens twice per layer, and additionally, splitting and concatenating the data into two streams incurs extra computational cost.
> >
> > Regarding the reviewer's concern about the apparent contradiction with VMamba’s results on multi-scan efficiency, our findings are actually not contradicted. The introduction of our auxiliary token swapping reverses the efficiency trend, making "uni-scan + swap" the most efficient configuration. In Table 3 of the paper, especially with the “MambaVision-T w/o Attention" setting, uni-scan alone achieves only 80.0\%, whereas parallel bi-scan improves accuracy to 81.1\%, showing that parallel bi-directional scanning is effective. However, uni-scan + swap reaches 81.0\% while being much faster, thus becoming the more efficient choice. In this way, the proposed auxiliary token swapping is a breakthrough.
> >
> > Additionally, our model uses Attention alongside Mamba, which partly explains the different trends we observe. VMamba conducted comparisons without leveraging recent acceleration techniques such as FlashAttention or FlashAttention-2, which likely resulted in an underestimation of Attention’s effectiveness. Our hybrid design delegates part of the bidirectional information flow to Attention, enabling state-of-the-art efficiency while using uni-scan.
> >
> > ## Paper Structure (W3)
> > We thank the reviewer for this suggestion. We have moved the MambaVision macro-architecture section into Appendix. The freed space is used to expand our analysis of downstream results, including more detailed quantitative and qualitative comparisons.
> >
> > ## Downstream Task Performance (W4)
> > We thank the reviewer for pointing this out. We realized that the way we visualized our results may have contributed to the impression that the performance gains on downstream tasks were limited. Our primary focus was to develop an efficient model that achieves a favorable accuracy–speed trade-off. As a result, when comparing within size categories such as tiny, small, and base, our model may have appeared to offer smaller gains simply because it is lighter than competing methods. However, as shown in the newly added **Fig. 5 and 9**, our accuracy–throughput trade-off has a clear improvement over existing methods on downstream tasks as well.

---

### Official Review · Reviewer_G5Nk · 2025-10-29

**Soundness:** 3
**Presentation:** 4
**Contribution:** 3
**Rating:** 4
**Confidence:** 4

**Summary:**

The paper presents SF-Mamba, an improved visual Mamba architecture for vision tasks. SF-Mamba introduces several innovations: auxiliary patch swapping with an extra token, which enables bidirectional information flow during a unidirectional scan, and batch folding with periodic state reset, which enhances GPU parallelism. Extensive experiments demonstrate that SF-Mamba significantly outperforms existing models in terms of accuracy and throughput across image classification and segmentation tasks.

**Strengths:**

1. The motivation is good. Previous methods addressed the problem from the perspective of multiple scans, whereas this paper innovatively addresses the Mamba architecture's issue of sequential reasoning by focusing on a single-scan approach for causal information swapping.

2. By restructuring the tensor dimensions of batch data, the paper improves parallelism from a GPU computation perspective, which benefits the acceleration of large-scale training.

3. The experiments are comprehensive, covering CNN architectures, transformer architectures, and hybrid architectures. Detailed ablation studies and supplementary materials provide a thorough exploration of the proposed method.

**Weaknesses:**

1. The performance improvement of the model is not significant. The accuracy improvement is not obvious compared to the baseline Mamvbavision. Moreover, the gain in speed is not so significant.

2. The comparison models do not include Fast R-CNN or Faster R-CNN for the object detection experiment, and the comparison in the validation set is insufficient.

3. The method of exchanging token positions to achieve contextual structure interaction may not be the optimal approach. The current ablation studies do not directly prove that the performance improvement is caused by the position swapping.

4. What about other hyperparameters designed in the results in Figure 4? Does it have an optimal design for performance and speed?

**Questions:**

Please see the weakness. I would raise the ratings if the rebuttal addressed the questions.

---

> ### Author Response · Authors · 2025-11-27
> **Reply to G5Nk (W1, W2, W3, W4)**
>
> We thank the reviewer for the thoughtful feedback and positive comments highlighting our motivation, GPU efficiency improvements, and the comprehensiveness of our experiments and ablations studies. In the following, we address the main concerns in detail.
>
> ## Significance of Accuracy and Speed Improvements (W1)
>
> While each gain over MambaVision may appear modest, we achieved higher accuracy while reducing computational cost. Based on Fig. 1, we believe that the combined accuracy-throughput tradeoff improvement is about as large as the progress this well-established field usually makes in an entire year. Moreover, in our revised version of the paper, we improved the implementation of the 1D convolution for batch-folded data, achieving slightly better speed than before.
>
> ## Comparison with Other Object Detection Baselines (W2)
> We appreciate the suggestion to extend comparisons to show the robustness of our method. We tried Faster R-CNN and Mask R-CNN on COCO using a common 1x (12 epoch) schedule training. Since recent methods no longer use Fast R-CNN and it has become difficult to compare fairly with state-of-the-art backbones, we conducted our experiments using Mask R-CNN instead. However, please let us know if you would prefer the experiments with Fast R-CNN.
>
> **[Detection on COCO dataset using Faster R-CNN.]**
> | Backbone | AP box | fps |
> |--------|--------|------|
> | RegNetX-3.2GF | 39.9 | 31.6 |
> | MambaVision-T | 42.4 | 37.2 |
> | SF-Mamba-T | 43.2 | 41.7 |
> |--------|--------|------|
> | MambaVision-S | 43.9 | 37.2 |
> | SF-Mamba-S | 44.9 | 40.8 |
> |--------|--------|------|
> | MambaVision-B | 46.2 | 30.0 |
> | SF-Mamba-B | 47.6 | 34.8 |
>
> **[Detection and instance segmentation on COCO dataset using Mask R-CNN.]**
> | Backbone | AP box | AP mask | fps |
> |--------|--------|------|------|
> | VMamba-T | 47.3 | 42.7 | 29.9 |
> | MambaVision-T | 43.1 | 40.0 | 35.5 |
> | SF-Mamba-T | 43.8 | 40.3 | 40.9 |
> |--------|--------|------|
> | VMamba-S | 48.7 | 43.7 | 23.3 |
> | MambaVision-S | 44.4 | 41.0 | 33.4 |
> | SF-Mamba-S | 45.3 | 41.5 | 41.0 |
> |--------|--------|------|
> | VMamba-B | 49.2 | 44.1 | 20.0 |
> | MambaVision-B | 46.7 | 42.8 | 29.8 |
> | SF-Mamba-B | 47.8 | 43.5 | 34.6 |
>
> These results show consistent accuracy and speed improvements of our method against MambaVision. Although our method is weaker in accuracy compared to VMamba within the same categories (T, S, or B), when comparing VMamba-**T** and SF-Mamba-**B**, SF-Mamba-B surpasses VMamba-T in both speed and accuracy, clearly demonstrating a superior performance–throughput trade-off. The newly added **Fig. 5 and 9** should be easy to understand the trade-off.
>
> ## Effectiveness of Token Swapping (W3)
> Thank you for the insightful comment. As you pointed out, our current ablation study did not fully demonstrate that the performance gains were indeed due to token swapping. To verify this more conclusively, we conducted an additional experiment in which we added two learnable tokens in the manner of class tokens used in ViT or Vim while removing the swapping mechanism. The results are shown below.
>
> **[Verification of the importance of swapping.]**
> | swap | auxiliary token | acc. | img/s |
> |--------|--------|------|------|
> |  |  | 82.2 | 7645 |
> |  | learnable | 82.1 | 7613 |
> | w/ (ours) | learnable | 82.3 | 7585 |
> | w/ (ours) | data-dependent (ours) | **82.4** | 7602 |
>
> Simply adding learnable tokens leads to degraded performance, indicating that the bi-directional information flow enabled by swapping plays a crucial role. We believe that including these results clearly demonstrates that the swapping operation is responsible for the performance improvements.
>
> We also added **Fig. 7** to the Appendix. This figure visualizes the ERF of the layers up to the Attention layer, showing that swapping indeed enables bidirectional context exchange under a uni-scan.
>
> ## Hyperparameter Sensitivity of Figure 4 (W4)
>
> There are no hyperparameters involved in determining the optimal $B_1$ value for batch folding in Fig. 4. We pre-search the optimal $B_1$ based on the input tensor shape [batch size, dimension, state dimension, sequence length], store the results in a lookup-table format, and then retrieve the optimal $B_1$ value from this table during inference or training.

---

### Official Review · Reviewer_V3Nn · 2025-10-30

**Soundness:** 3
**Presentation:** 3
**Contribution:** 3
**Rating:** 6
**Confidence:** 5

**Summary:**

This paper proposes SF-Mamba, a novel visual encoder addressing limitations of existing Vision Transformers and visual Mamba models. ViTs suffer from quadratic complexity, while visual Mamba faces non-causal information flow constraints and inefficiency with short tokens. SF-Mamba introduces two core innovations: auxiliary patch swapping for bidirectional information flow under unidirectional scanning, and batch folding with periodic state reset to enhance GPU parallelism. Extensive experiments on image classification, object detection, and segmentation demonstrate superior accuracy-throughput trade-offs compared to SOTA baselines.

**Strengths:**

1. This paper targets critical pain points of visual Mamba (causality constraint, short-sequence inefficiency) with lightweight, non-intrusive solutions.
2. Comprehensive validation across three fundamental vision tasks (classification, detection, segmentation) with consistent performance gains.
3. Practical optimizations (e.g., adaptive $\(B_1\)$, Triton kernel for swapping) enhance real-world applicability, with code release planned.

**Weaknesses:**

1. The macro-architecture heavily relies on MambaVision’s hybrid (Mamba+Attention) design, lacking significant innovations in overall network structure.
2. Ablation studies on auxiliary token initialization and discard timing are limited; deeper analysis of their impact on different tasks is needed.
3. No discussion on generalization to ultra-high-resolution images or low-resource devices (e.g., edge GPUs), restricting scope insights.

**Questions:**

Please refer to the weakness part.

---

> ### Author Response · Authors · 2025-11-27
> **Reply to V3Nn (W1, W2)**
>
> We sincerely thank the reviewer for the positive evaluation and constructive comments. We are pleased that the reviewer recognized the practical motivation of our work, the consistent improvements across three fundamental tasks, and our practical implementation. Below, we address the remaining concerns.
>
> ## Dependence on MambaVision Architecture (W1)
>
> We appreciate the reviewer’s observation. Indeed, our work builds on the hybrid Mamba–Attention backbone proposed in MambaVision. This choice is intentional: our goal is not to redesign the macro-architecture, but rather to improve the core efficiency and information flow of state-space modules, which is a complementary and orthogonal direction.
> The two core contributions--auxiliary patch swapping and batch folding with periodic reset--can be plugged into other visual Mamba variants. To demonstrate this, we have added experiments on Vim architecture in **Table 11 and Appendix D.6**.
>
> **[Comparison on Vim macro-architecture]**
> | size | scan | Params | MACs | img/s | acc. |
> |--------|--------|------|------|------|------|
> | S | parallel-bi scan (Vim) | 26M | 5.3G | 1079 | 80.3 |
> | S | uni-scan | 26M | 4.9G | 1639 | 79.3 |
> | S | uni-scan + swap (ours) | 26M | 5.0G | 1614 | 80.1|
> | S | uni-scan + swap + Bfold (ours) | 26M | 5.0G | **2022** | 80.1 |
> |--------|--------|------|------|------|------|
> | T | parallel-bi scan (Vim) | 7M | 1.5G | 2094 | 76.3 |
>
> When we remove the modules responsible for the inverse-direction scan in the Vim-S architecture and make the model uni-directional, the accuracy drops, but the speed improves significantly. By introducing the proposed auxiliary token swapping, we can recover most of the lost accuracy while preserving the improved speed. The resulting model achieves a speed similar to Vim-T but with substantially better accuracy than Vim-T, demonstrating that our method offers a clearly superior accuracy–throughput trade-off.
>
> ## Ablation Study on Downstream Task (W2)
> We thank the reviewer for this valuable suggestion. Following the comment, we have expanded our ablation studies by evaluating auxiliary token initialization and discard timing on the **ADE20K segmentation task**, which provides a complementary perspective beyond classification. These results have been added to **Table 2**.
>
> **[Ablation studies about auxiliary token swapping  on ADE20K segmentation task.]**
> | swap | auxiliary token | discard timing | mIoU |
> |--------|--------|------|------|
> |  |  |  | 46.0 |
> | w/ | learnable | before attn | 46.5 |
> | w/ | data-dependent | before attn | 46.8 |
> | w/ | data-dependent | after 1st attn | **47.2** |
> | w/ | data-dependent | after attn | 46.6 |
>
> The trends are almost consistent with those in the classification experiments.
> We hope these additional results provide the deeper analysis the reviewer was seeking. If the reviewer has a particular direction or a specific aspect they would like to see further explored, we would be very happy to investigate it.

---

> ### Author Response · Authors · 2025-11-27
> **Reply to V3Nn (W3)**
>
> # High-resolution and Edge Scenarios (W3)
>
> Thank you for pointing out this feedback. As suggested, we conducted a throughput comparison at higher input resolutions, and the results have been included in **Table 12 and Appendix D.7**. These results show that our proposed improvements preserve their benefits even as resolution scales, and that the strategy using windowed Attention while using Mamba globally (SF-Mamba♣) remains particularly robust under extremely large inputs.  This may reduce training and inference costs in high-resolution applications such as medical imaging, aerial monitoring, and robotics.
> **[Throughput [img/s] comparison with different resolutions.]**
> | resolution |224 | 448 | 896 | 1792 | 3584 |
> |--------|--------|------|------|------|------|
> | VMamba-T (bs=32) | 1384 | 402 | 107 | 5 | OOM |
> | FasterViT-0 (bs=32) | 1415 | 1400 | 418 | 99 | OOM |
> | MambaVision-T (bs=32) | 3770 | 1578 | 324 | 50 | 5 |
> | SF-Mamba-T (bs=32) | **3962** | **1777** | 397 | 61 | 6 |
> | SF-Mamba$\clubsuit$-T (bs=32) | - | - | **427** | **105** | **27** |
> |--------|--------|------|------|------|------|
> | VMamba-T (bs=1) | 62 | 62 | 62 | 24 | 5 |
> | FasterViT-0 (bs=1) | 44 | 43 | 43| 43 | 12 |
> | MambaVision-T (bs=1) | 119 | 121 | 120 | 48 | 5 |
> | SF-Mamba-T (bs=1) | **126** | **126** | **125** | 54 | 6 |
> | SF-Mamba$\clubsuit$-T (bs=1) | - | - | 120 | **89** | **26** |
>
> We acknowledge that we were unable to include edge-GPU or mobile-device benchmarks in this submission due to the lack of hardware access and the required engineering effort to adapt the kernel implementations.
> Importantly, previous works, e.g., [1, 2], has shown that Mamba kernels can be deployed on Jetson (via TensorRT) and also on iOS devices, suggesting that our method should likewise be portable. We have added a Broader Impact section (**Appendix A**) in the manuscript outlining the steps needed for adapting our kernels to such platforms and why the proposed batch-folding and swapping strategies are compatible with existing mobile-Mamba pipelines.
>
> [1] https://github.com/s990093/Mamba-Orin-Nano-Custom-S6-CUDA
>
> [2] TensorRT LLM, https://github.com/NVIDIA/TensorRT-LLM.

---

### Official Review · Reviewer_MDas · 2025-10-31

**Soundness:** 2
**Presentation:** 3
**Contribution:** 2
**Rating:** 4
**Confidence:** 4

**Summary:**

This manuscript describes a modification to vision state space models. To improve the processing speed of mamba-based vision models, the authors proposed to 1) auxilliary patch swapping for bidirectional information flow, and 2) batch folding with periodic state reset. Experiments shows positive results on these components.

**Strengths:**

+ The manuscript is well-presented and easy to follow.
+ It is good to see analysis on the inference speed of the mamba based model.
+ The reset trick is interesting.

**Weaknesses:**

+ The proposed method is only testifed on MambaVision. It should be applied to more mamba-based models to support the claim of "Rethinking State Space Model for Vision".
+ Swapping last token is not equivant to bi-directional scan, and the author failed to prove the superiority of swapping last token. As in table 3, if the attention is removed, swapping last token worse than parallel bi-scan and even series bi-scan. Since attention itself is a undirectional operation, this seems that switching to swapping last token is not working but attention works.

**Questions:**

+ In the manuscript of MambaVision, they utilized the same hardware, but they reported a throughput of  3670 img/s for MambaVision-B. However, in this manuscript, the speed is downgraded to 2974 img/s. Why? If jittering exists, please report mean and std over multiple test runs.
+  Why Mamba kernel requires 32 parallel threads? If current mamba kernel is not suitable for short sequence of image classification, why do we need parallel scan?  Why do we need mamba? The performance drops severly in Tab.3 if attention are removed.
+ In Table 3, why the inference troughput drop when attention is removed? If that so, the inference scheme may not be appropriate. Please report performance under large batch size, say, 2048 as in SHViT and EfficientViT.
+  Can the proposed method be applied to other mamba-based model, say, Vim, VMamba and their variants?

---

> ### Author Response · Authors · 2025-11-27
> **Reply to MDaS (W1, Q4, W2)**
>
> We thank the reviewer for the detailed and constructive feedback. We appreciate the positive remarks regarding our presentation quality, the analysis of the inference speed, and the interest in the batch folding mechanism. Below, we address each concern.
>
> ## Generalization to other Mamba-based models (W1, Q4)
> We agree that validating the proposed components on other Mamba-based vision models would strengthen our claim. Therefore, we additionally evaluated the performance using the Vim architecture as follows.
> We have included the discussions and results to **Table 11 and Appendix D.6**.
>
> **[Comparison on Vim macro-architecture]**
> | size | scan | Params | MACs | img/s | acc. |
> |--------|--------|------|------|------|------|
> | S | parallel-bi scan (Vim) | 26M | 5.3G | 1079 | 80.3 |
> | S | uni-scan | 26M | 4.9G | 1639 | 79.3 |
> | S | uni-scan + swap (Ours) | 26M | 5.0G | 1614 | 80.1|
> | S | uni-scan + swap + Bfold (Ours) | 26M | 5.0G | **2022** | 80.1 |
> |--------|--------|------|------|------|------|
> | T | parallel-bi scan (Vim) | 7M | 1.5G | 2094 | 76.3 |
>
> When we remove the modules responsible for the inverse-direction scan in the Vim-S architecture and make the model uni-directional, the accuracy drops, but the speed improves significantly. By introducing the proposed auxiliary token swapping, we can recover part of the lost accuracy—though not fully up to the original Vim-S level—while preserving the improved speed. The resulting model achieves a speed similar to Vim-T but with substantially better performance than Vim-T, demonstrating that our method offers a clearly superior accuracy–throughput trade-off.
>
> Unfortunately, our auxiliary token swapping cannot be applied to methods like VMamba that heavily rely on 2D convolutions in each Mamba block. This is because adding the auxiliary tokens prevents us from reshaping them back into the original 2D layout, making it impossible to apply 2D convolutions. Although our method cannot be applied to VMamba, Fig. 1 of the main paper shows that our approach significantly surpasses VMamba in terms of accuracy–throughput trade-off, and thus we believe that the following conclusions of our “rethinking SSM for Vision” remain firmly valid.
>
>
> ## Role of Token Swapping and Attention (W2)
> Thank you for this insightful comment!
> We have expanded our ablation studies and now report the Attention-only results in **Table 14 and Appendix D.8** for a clearer comparison.
>
> **<Attention in our architecture is important, but SSM (w/ swap) is also essential.>**
> | Arch. | Params | img/s | acc. |
> |-------|------|------|------|
> | Attention only | 34.2M | 7803 | 82.3\% |
> | SSM only | 29.4M | 7395 | 80.2\% |
> | SSM only (w/ swap) | 29.4M | 7306 | 81.0\% |
> | Hybrid | 31.8M | 7645 | 82.2\% |
> | Hybrid (w/ swap) | 31.8M | 7600 | **82.5\%** |
>
> These results show that while Attention provides beneficial bidirectional information flow, it alone is not sufficient to match the full hybrid model. Conversely, SSM alone lags behind, but incorporating the **swap mechanism** yields a clear improvement. The best performance is achieved only when *both* components--Attention and SSM (with swap)--are present. This supports our claim that token swapping plays a complementary role to Attention rather than replacing it.
>
> **Clarification regarding Table 3 in the main paper**. It appears there may have been a misunderstanding in the interpretation of Table 3. Under MambaVision-T (w/o Attention), **our uni-scan + swap** variant (81.0\%) outperforms the series bi-scan variant (80.4\%). The parallel bi-scan variant achieves slightly higher accuracy (81.1\%), but at the cost of a notable throughput drop. This indicates that parallel bi-scan is less efficient in practice.
> Thus, uni-scan + swap offers a better accuracy-efficiency trade-off, and scaling this design is a more practical strategy.

---

> ### Author Response · Authors · 2025-11-27
> **Reply to MDaS (Q1, Q2, Q3)**
>
> ## MambaVision-B Throughput Discrepancy (Q1)
> Thank you for highlighting this discrepancy. We carefully re-examined our benchmarking setup to ensure fairness and consistency. We used essentially the same software environment as the original MambaVision implementation, with one minor difference: cuDNN was upgraded from v8 to v9, which slightly accelerates Transformer layers. This explains why our measured throughputs for **MambaVision-T/S** are marginally *higher* than those reported in the original paper.
>
> In contrast, the speed of MambaVision-B is indeed lower than the values reported in the original paper. However, our inference-time measurements are highly stable with a 500 run average. We checked the throughput 10 times, resulting in 2974 ± 9 [img/s]. Therefore, we are confident that our reported number is accurate and stable rather than the result of randomness or misconfiguration.
>
> We also note that, unlike the T/S variants in their original paper, MambaVision-B incorporates additional components (e.g., LayerScale) to boost accuracy. Based on our investigation, we suspect that the original paper may have reported throughput using an earlier, pre-improved version of the architecture—before these heavier modules were enabled—leading to the observed discrepancy.
>
>
> ## Necessity of 32 Parallel Threads and the necessity of Mamba (Q2)
> **Why 32 threads?** The requirement is not specific to Mamba--it comes from **GPU hardware constraints**. Modern NVIDIA GPUs execute instructions in units called warps, each consisting of 32 threads. Even if a kernel is launched with fewer than 32 threads, the hardware still allocates a full warp, meaning that execution effectively happens with 32 threads internally. Thus, implementing warp-scan and related kernels using 32-thread groups is both natural and efficient.
> For AMD GPUs, the analogous execution group (wavefront) is 64 threads, which is why ROCm’s warp-scan implementation uses 64-thread units.
> This is a hardware-level constraint, not a limitation of our algorithm or implementation.
>
> **Why do we need parallel scan for short sequences?** Even with inefficient use of GPU threads, parallel execution is still considerably faster, so using a parallel scan remains beneficial (please refer to "scan ours" and "scan Pytorch" in Fig. 8 of the original Mamba paper). Moreover, the inefficiency problem can be solved with our batch folding.
>
> **Why do we need Mamba?** The motivation for using Mamba instead of pure Attention is that combining Mamba with Attention in a hybrid manner improves classification accuracy and  enables substantial computational savings when handling high-resolution inputs in downstream tasks such as segmentation and object detection, as empirically shown in **Fig. 5 and 9**.
>
> [1] https://rocmdocs.amd.com/projects/rocPRIM/en/develop/warp_ops/scan.html
>
> # Throughput Drop without Attention (Q3)
>
> The reason why the attention-only architecture is faster is that, with a 224×224 input, the token lengths used in stage 3 (15×15) and stage 4 (7×7) are short enough that Mamba remains slower even when accelerated with our batch folding. However, our method still offers advantages by delivering higher accuracy, even if it is slightly slower.
>
> Based on the reviewer's suggestion, we investigated how our method behaves under different batch sizes as follows. We have included these results into **Table 13 and Appendix D.7**.
>
> **[Throughput with different batch sizes]**
> | Arch. | 1 | 32 | 128 | 256 | 512 | 1024 |
> |-------|----|----|----|----|----|----|
> | MambaVision-T | 119 | 3770 | 6662 | 7025 | 7134 | 7271 |
> | ours-T | **126** | **3962** | **7600** | **7801** | **8009** | **8190** |
> |-------|----|----|----|----|----|----|
> | MambaVision-B | 96 | 2798 | 2974 | 3128 | 3176 | 3206 |
> | ours-B | **98** | **3168** | **3534** | **3592** | **3641** | **3685** |
>
> Although auxiliary token swapping introduces a slight increase in computational cost, the results show that throughput consistently improves thanks to batch folding and other optimizations.

---

### Author Response · Authors · 2025-11-27
**Dear all reviewers and new area chairs**

# Summary of Revisions
We sincerely appreciate the valuable feedback provided by the reviewers. We are convinced that incorporating their suggestions has substantially enhanced the quality of our paper, and **we have accordingly included many additional experiments and discussions.**  The following is a summary of the revision in the paper. **We believe following additional information which have already included in the paper will solve all concerns from reviewers.**

- Add visualization of auxiliary token behavior with swapping (**Fig. 7**).
- Extend ablation studies on auxiliary token behavior with swapping (**Table 2**).
- Add figures to show the effectiveness in downstream tasks (**Fig. 5 and 9(a)**)
- Add experiments adding our method on Vim macro-architecture to demonstrate generalization (**Table 11**).
- Add Faster R-CNN and Mask R-CNN downstream task results to show the robustness and comparison with other methods (**Table 9 and 10, Fig. 9(b)**).
- Extend ablation studies about scan operations (**Table 3**)
- Add analysis about high-resolution and different batch sizes (**Table 12 and 13**).
- Add "Broader Impacts" section to discuss potential benefits.
- We improved our Conv1D implementation for batch folded data, and our inference speed becomes even faster (All throughput and fps with batch folding).

---

### Meta-Review · Area_Chair_zbDM · 2025-12-27

**Summary:**

The reviewers raised major concerns about contributions of the proposed model and performance improvement. Particularly, they pointed out unclear evaluation of the attention module and marginal performance improvements over compared methods. The authors provided detailed responses, but did not resolve the concerns raised by reviewers.  The authors are suggested to carefully consider the concerns raised by reviewers and submit their work to the next conference.

**Reviewer Concerns:**

The authors provided detailed responses, but did not resolve the concerns raised by reviewers.

**Reviewer Scores:**

The reviewers who assigned negative scores did not provide further discussion. There is no evidence to show the reviewers would have changed their score if they had been able to participate fully in the discussion.

---

### Decision · Program_Chairs · 2026-01-26

Reject